# Proteomic and functional analyses of the periodic membrane skeleton in neurons

Ruobo Zhou [1,2,3,4,8 ✉], Boran Han [1,2,3,8], Roberta Nowak[5], Yunzhe Lu[6], Evan Heller[1,2,3], Chenglong Xia [1,2,3], Athar H. Chishti[6], Velia M. Fowler [5,7] & Xiaowei Zhuang [1,2,3 ✉]

Actin, spectrin, and associated molecules form a membrane-associated periodic skeleton (MPS) in neurons. The molecular composition and functions of the MPS remain incompletely understood. Here, using co-immunoprecipitation and mass spectrometry, we identified hundreds of potential candidate MPS-interacting proteins that span diverse functional categories. We examined representative proteins in several of these categories using super-resolution imaging, including previously unknown MPS structural components, as well as motor proteins, cell adhesion molecules, ion channels, and signaling proteins, and observed periodic distributions characteristic of the MPS along the neurites for ~20 proteins. Genetic perturbations of the MPS and its interacting proteins further suggested functional roles of the MPS in axon-axon and axon-dendrite interactions and in axon diameter regulation, and implicated the involvement of MPS interactions with cell adhesion molecules and non-muscle myosin in these roles. These results provide insights into the interactome of the MPS and suggest previously unknown functions of the MPS in neurons.

[1] Howard Hughes Medical Institute, Harvard University, Cambridge, MA 02138, USA. [2] Department of Chemistry and Chemical Biology, Harvard University, Cambridge, MA 02138, USA. [3] Department of Physics, Harvard University, Cambridge, MA 02138, USA. [4] Department of Chemistry, The Pennsylvania State University, University Park, PA 16802, USA. [5] Department of Molecular Medicine, The Scripps Research Institute, La Jolla, CA 92307, USA. [6] Department of Developmental, Molecular, and Chemical Biology, Tufts University School of Medicine, Boston, MA 02111, USA. [7] Department of Biological Sciences, The University of Delaware, Newark, DE 19716, USA. [8] These authors contributed equally: Ruobo Zhou, Boran Han. ✉email: ruobo.zhou@psu.edu; zhuang@chemistry.harvard.edu

Super-resolution imaging has recently revealed a membrane-associated periodic skeleton (MPS) in neurons[1]. The neuronal MPS contains some structural components that are homologous to the membrane skeleton of erythrocytes[2,3], but adopts a distinct ultrastructure, in which actin filaments are assembled into ring-like structures, and the adjacent actin rings are connected by spectrin tetramers consisting of two αII-spectrin and two β-spectrin (βII, βIII or βIV) subunits, forming a one-dimensional (1D) periodic structure with an ~190-nm period underneath the plasma membrane of neurites[1]. In mature neurons, the MPS structure spans the entire axonal shaft, including both the axon initial segment and distal axon[1,4,5], and has also been observed in a substantial fraction of dendritic regions[4,6,7]. In soma and a fraction of dendritic regions, a 2D polygonal structure resembling the membrane skeleton of erythrocytes has been observed[7], but the average end-to-end distance of the spectrin tetramer connecting the two adjacent actin nodes in the native environment is significantly greater in neurons than in erythrocytes[7,8]. A highly prevalent structure in the nervous system, the MPS is present in diverse neuronal types, including excitatory and inhibitory neurons in both central and peripheral nervous systems[9,10], and across diverse animal species, ranging from C elegans to humans[10]. This submembrane lattice structure can organize transmembrane proteins[1,4,6,11–13], and it has been shown recently that the MPS can function as a signaling platform that coordinates the interactions of key signaling proteins and enables signal transduction in neurons[13]. Additional functional roles of the MPS have also been demonstrated or implicated in axon stability under mechanical stress[1,14], mechanosensation[15], diffusion restriction at the AIS[16], and axon degeneration[17,18]. Disruption of the MPS causes a range of neurological impairments in mice[19,20].

Despite the importance of the MPS in neurons, its molecular components and interacting partners have not been systemically investigated, which limits our understanding of the functions of the MPS and the molecular mechanisms underlying these functions. In addition to actin and spectrin, two other structural components of the MPS have been identified: adducin, a protein that caps the fast-growing end of actin filaments, and ankyrin, an adaptor protein that can anchor transmembrane proteins to the MPS[1,4,5]. A few transmembrane proteins have also been observed to associate with the MPS structures in the AIS region, including ion channels and adhesion molecules[1,4,6,11,12]. Recently, it has been shown that several transmembrane signaling proteins, including the G-protein coupled receptor CB1, cell adhesion molecule NCAM1, and two receptor tyrosine kinases (RTKs), TrkB and FGFR, can be recruited to the MPS structure in response to extracellular stimuli to enable RTK transactivation in neurons[13]. The list of MPS components and interacting partners is, however, likely far from complete.

In this work, we used co-immunoprecipitation (co-IP) to pull down proteins directly or indirectly associated with the MPS from cultured mouse hippocampal neurons and adult mouse brains, followed by mass spectrometry for a proteomic-scale identification of the candidate structural components and interacting partners of the MPS. This analysis revealed hundreds of potential candidate MPS-interacting proteins that directly or indirectly associate with the MPS and these proteins span many functional categories and cellular pathways. In addition, we used quantitative mass spectrometry to investigate how the expression levels of proteins are differentially regulated at the proteomic scale upon disruption of the MPS. We further used stochastic optical reconstruction microscopy (STORM)[21,22], a super-resolution imaging method, to examine a subset of the candidate proteins in several important functional categories, including actin-binding proteins, motor proteins, cell adhesion molecules, ion channels, and other signal transduction proteins, and observed periodic distributions characteristic of the MPS for ~20 proteins, suggesting that they are associated directly or indirectly with the MPS. Using genetic knockout or shRNA knockdown, we identified several structural proteins that are essential for the MPS formation. Our genetic perturbation experiments further suggested functional roles of the MPS in axon-axon and axon-dendrite interactions, as well as in axon diameter regulation.

## Results

**Proteomic analysis of candidate structural components and interacting partners of the MPS.** We first used co-IP and mass spectrometry to pull down and identify candidate structural components and interacting partners of the MPS from cultured neurons (Fig. 1a). Magnetic beads were coated with the antibody that can specifically bind to a bait protein known to be an MPS structural component, including βII-spectrin, αII-spectrin and α-adducin[1], and these beads were then incubated with the lysate from cultured mouse hippocampal neurons at 20 days in vitro (DIV) to allow capturing of the proteins that bind directly or indirectly to these known MPS components. The co-immunoprecipitated protein mixtures were subsequently analyzed using SDS-PAGE and mass spectrometry separately.

The co-immunoprecipitated proteins showed notable enrichment of proteins at the expected molecular weights of spectrin and actin, as well as other proteins, in the SDS-PAGE gel images, whereas co-IP using beads coated with a control IgG hardly pulled down anything (Supplementary Fig. 1). Furthermore, treatment with actin disrupting drugs (LatA and CytoD), which are known to disrupt the MPS[1,4,7], substantially reduced the amount of co-immunoprecipitated proteins (Supplementary Fig. 1).

Next, we used mass spectrometry to identify the co-immunoprecipitated proteins. To reduce the number of potential false positives of the identified MPS structural components and interacting proteins, we only considered the proteins commonly identified using all three different bait proteins (βII-spectrin, αII-spectrin, and α-adducin), although this criterion may filter out some of the true MPS-interacting proteins. This resulted in a list of 480 proteins (Fig. 1b and Supplementary Data 1). Actin, βII-spectrin, α-adducin and ankyrin, which have been previously shown to be structural components of the neuronal MPS[1], are indeed among the 480 proteins identified here. It is worth noting that because the lysates used in the co-IP experiments were not specific to the fraction of any specific neuronal sub-compartment, the identified candidate protein list could thus include structural components and interacting partners of the membrane skeleton structures in axons, dendrites and soma. As a cautionary note, as is typically true for proteomic analysis by co-IP and mass spectrometry, false positives may arise from non-specific protein binding to antibody-coated beads. Although we have performed negative controls with the IgG-coated beads in order to remove these false positives (Methods), such removal may not be complete. It is also known that liquid chromatography mass spectrometry (LS-MS) can suffer from run-to-run variations, as we observed between replicates. Thus, the proteins included in this list are potential candidates of structural components and interacting partners of the MPS that need to be further validated, as we did for a subset of these candidates shown in later sections. Furthermore, by interaction partners, we refer to proteins that either directly interact with the MPS or are indirectly associated with the MPS.

For global analysis of these identified candidate proteins, we performed gene ontology (GO) term analysis of the biological process (BP) category using the DAVID platform[23]. The functional annotation clusters of enriched GO BP terms

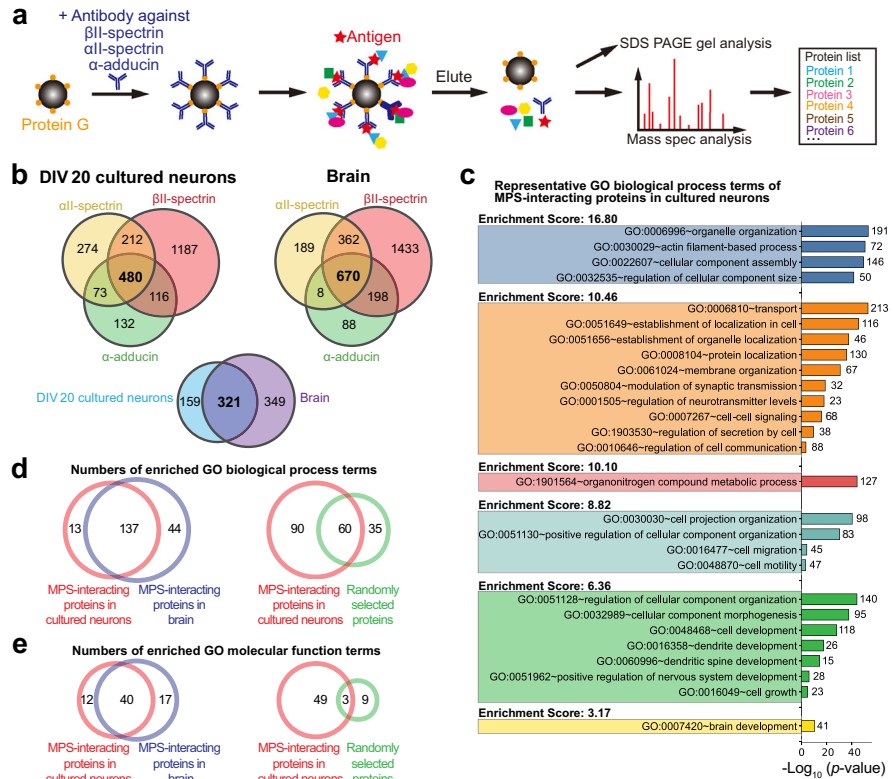

**Fig. 1 Proteomic analysis and GO term enrichment analysis of candidate MPS-interacting proteins. a** Schematic for co-immunoprecipitation (co-IP)-based mass spectrometry identification of candidate MPS-interacting proteins. The antibody against a bait protein, βII-spectrin, αII-spectrin, or α-adducin, was attached to protein-G-coated beads. The antibody-coated beads were then used to capture the bait protein and the co-immunoprecipitated proteins from the cultured hippocampal neuron lysate or adult mouse whole-brain lysate. The co-immunoprecipitated proteins were identified using mass spectrometry. **b** Top: Venn diagrams showing the overlap of the identified proteins in three co-IP experiments using βII-spectrin, αII-spectrin or α-adducin as the bait, either from cultured hippocampal neuron (DIV 20) lysates (left) or from mouse whole-brain lysates (right). Two biological replicates were performed for each co-IP condition. Bottom: Venn diagram showing the overlap between the 480 identified proteins from cultured hippocampal neurons and the 670 identified proteins from mouse whole-brain lysates (See Supplementary Data 1 for the full list of the candidate MPS-interacting proteins). **c** Functional annotation clustering of the enriched Gene Ontology (GO) terms in the biological process (BP) category for the 480 candidate MPS-interacting proteins identified in cultured hippocampal neurons. A selected subset of GO BP terms are shown (See Supplementary Data 2 for the full list of clustered GO BP terms enriched in the candidate MPS-interacting proteins). The corresponding p-values (bars) and protein numbers (next to the bars) for each enriched GO BP term are shown on the right. **d** Venn diagrams showing the overlap between the enriched GO BP terms of the 480 candidate MPS-interacting proteins identified in cultured mouse hippocampal neurons (DIV 20) and the enriched GO BP terms of the 670 candidate MPS-interacting proteins identified in the adult mouse whole brain (left). Also shown in comparison is the overlap between the enriched GO BP terms of the 480 candidate MPS-interacting proteins in cultured mouse hippocampal neurons (DIV 20) and the enriched GO BP terms of 670 randomly selected genes from the mouse genome (right). **e** Same as **d** but for GO MF terms instead of GO BP terms.

(p-value < 0.05) showed the potential relationship of the MPS to many biological processes (Fig. 1c and Supplementary Data 2), including actin filament-based process, organelle organization and localization, protein localization, membrane organization, modulation of synaptic transmission, cell-cell signaling, cell secretion and communication, organonitrogen compound metabolic process, cell projection organization, cell migration and motility, dendrite and dendritic spine development, cell development and growth, etc. Similar GO term analysis of the molecular function (MF) category also suggested potential involvement of the MPS in diverse cellular functions (Supplementary Fig. 2a and Supplementary Data 3).

Next, we repeated our co-IP experiments using the lysates from whole adult mouse brains. We identified 670 proteins that were commonly pulled down using all three different bait proteins, βII-spectrin, αII-spectrin, and α-adducin (Fig. 1b and Supplementary Data 1). The majority (321 of 480) of the candidate MPS-interacting proteins identified from cultured hippocampal neurons were also identified here from whole brain lysates. The greater number of proteins identified from the whole brain lysates

is not surprising, given the inclusion of not only hippocampal neurons but also neurons from other brain regions as well as non-neuronal cells in the whole brain lysates. The enriched GO terms of the 670 proteins identified from brain tissues largely overlapped with the enriched GO terms of the 480 proteins identified from cultured hippocampal neurons (Fig. 1d, e, and Supplementary Fig. 2b, c).

Next, we used multiplexed quantitative mass spectrometry based on Tandem Mass Tag (TMT) isobaric labeling[24] to systematically determine the protein abundance changes in the cultured hippocampal neurons upon βII-spectrin knockdown (Supplementary Fig. 3a), which is known to disrupt the MPS[4,7]. Among the proteins detected in our quantitative mass spectrometry experiment, 1347 showed statistically significant up or down regulation in βII-spectrin knockdown neurons as compared to control neurons treated with scrambled shRNA (Supplementary Data 4). Notably, although we do not expect all MPS-interacting proteins to be up or downregulated upon MPS disruption, nor do we expect all differentially expressed proteins to necessarily interact with the MPS, we found that the enriched

GO terms of the candidate MPS-interacting proteins determined by co-IP and mass spectrometry overlapped substantially with the enriched GO terms of the differentially expressed proteins upon MPS disruption (Supplementary Fig. 3b, c; Supplementary Data 5 and 6), suggesting that proteins probed in these two experiments likely function in common or related biological processes.

**Examination of candidate MPS structural components using super-resolution imaging.** The co-IP-based mass spectrometry analysis provided a list of candidate MPS-interacting proteins, not all of which are necessarily associated with the MPS. Additional experiments are needed to validate the direct or indirect association of these proteins with the MPS. Among the identified candidate MPS-interacting proteins, a notable number of them were actin-binding proteins, which potentially bind to the actin filaments in the MPS and hence regulate the stability of the MPS. To test whether these candidate proteins are associated with the MPS, we imaged them using STORM[21,22] to examine whether these proteins exhibit periodic distributions along axons similar to actin, spectrin and α-adducin as previously reported[1].

To visualize these proteins, we used either immunostaining with antibody against the endogenous proteins, or moderate expression of GFP-fusion proteins through low-titer lentiviral transfection followed by immunolabeling using anti-GFP antibodies. For the proteins that have high-quality antibodies, the first approach generally provided higher quality images, as indicated by a higher degree of periodicity of the protein distribution along axons for MPS structural components (Supplementary Fig. 4a), potentially because the GFP-fusion proteins cannot be as efficiently incorporated into the MPS due to perturbation of the GFP tag, overexpression, or competition with endogenous proteins. Indeed, expressing GFP-fusion proteins in neurons where the endogenous proteins were depleted can improve the incorporation efficiency of the fusion proteins into the MPS (Supplementary Fig. 4b). It is also worth noting that antibody quality for immunolabeling can vary substantially, even for the same target protein (Supplementary Fig. 4c). We therefore typically screened multiple antibodies to identify the highest quality antibody for imaging and when antibodies with adequate quality did not exist, we used the GFP-fusion approach.

Using these labeling strategies, we imaged five proteins that were previously known as actin or βII-spectrin binding proteins, including αII-spectrin, tropomodulin 1, tropomodulin 2, dematin, and coronin 2B. αII-spectrin and βII-spectrin (or βIII and βIV-spectrin) form spectrin tetramers that are expected to be a structural component of the MPS. We thus expect αII-spectrin to exhibit periodic distributions in the axons. Indeed, αII-spectrin has been previously shown to be periodically distributed in the AIS region[19], and here we showed that αII-spectrin also exhibited periodic distributions in axon regions outside the AIS (Fig. 2a). In addition to αII-spectrin, tropomodulin 1, tropomodulin 2, dematin, and coronin 2B also exhibited periodic distributions along the axons (Fig. 2b–e). To quantify the periodic distribution, we calculated for each imaged protein the average 1D autocorrelation function and its amplitude from randomly selected axon regions, which provides a quantitative measure of the periodicity of the measured distribution. We found that all five examined proteins showed periodic autocorrelation with a period of ~190 nm but with varying amplitudes (Fig. 2a–e), whereas such periodic autocorrelation was not observed in several negative controls including both membrane and cytosolic proteins (Supplementary Fig. 5). Two-color STORM imaging further showed that the SH3 domain of αII-spectrin (which is expected to be roughly at the center of the spectrin tetramer) and adducin (which is expected to bind to actin) shows alternating periodic patterns with a 180-degree phase shift (Supplementary Fig. 6a), consistent with notion that the MPS is made of actin rings connected by spectrin tetramers. Likewise, using two-color STORM, we observed that tropomodulin 1, tropomodulin 2, dematin, and coronin 2B exhibited periodic distributions that was 180-degree phase-shifted from the periodic distribution of the spectrin tetramer center, marked by C-terminus of the βII-spectrin (Supplementary Fig. 6b–e), suggesting that tropomodulin 1, tropomodulin 2, dematin, and coronin 2B are associated with the actin rings. These results suggest several previously unknown structural components for the MPS, in addition to the previously identified components, actin, spectrin, α-adducin, and ankyrin.

**Proteins essential for the MPS formation or maintenance.** To test whether these structural proteins are essential for MPS formation and/or maintenance in neurons, we depleted these proteins using either shRNA knockdown (Supplementary Fig. 7a) or knockout mice and quantified the integrity of the MPS under each genetic perturbation condition by measuring the degree of periodicity of the βII-spectrin distribution in axons, as indicated by the average autocorrelation function amplitude. Compared to the wild type (WT) neurons, the MPS was disrupted or partially disrupted in the neurons depleted of αII-spectrin, α-adducin, tropomodulin 1, or dematin, whereas depletion of ankyrin B, coronin 2B, or tropomodulin 2 did not significantly disrupt the MPS (Fig. 2f and Supplementary Fig. 7b). Our observation that αII-spectrin is essential for the MPS formation in axons is consistent with previous results that the MPS is disrupted in βII-spectrin knockdown neurons[4,7] and that αII-spectrin knockout disrupts the MPS in the AIS region[19]. Our observation that the MPS is partially disrupted in the α-adducin depleted mouse neurons is consistent with the previous result reported in *Drosophila* neurons depleted of the *Drosophila* adducin homolog[25]. It has been also reported that the axons in α-adducin knockout mouse neurons showed enlarged diameters, although the level of MPS disruption was not quantitatively assessed in this previous study[26]. In addition, we observed that disruption of the MPS by knocking down βII-spectrin or α-adducin also enlarged the average diameter of the dendrites (Supplementary Fig. 8).

In addition to examining which structural proteins are essential for the MPS formation/maintenance, for βII-spectrin, a known key component of the MPS[4,7], we further examined the role of its various protein domains. βII-spectrin contains the N-terminal CH domain responsible for actin binding and the C-terminal pleckstrin homology (PH) domain responsible for binding to PI(4,5)P$_2$ phosphatidylinositol lipids as well as some membrane proteins (Fig. 2g)[2]. We thus examined the effect of deleting the CH or PH domain on the MPS integrity. We expressed GFP-tagged full length βII-spectrin, or GFP-tagged βII-spectrin mutants with either CH or PH domain deleted (βII-spectrin-ΔCH or βII-spectrin-ΔPH), in neurons cultured from brain-specific βII-spectrin knockout mice, which we generated by crossing *βII-Spec^{flox/flox}* mice[27] with the Nestin-Cre mouse line. We found that only the full length βII-spectrin could support the MPS formation, exhibiting a periodic distribution along axons, whereas βII-spectrin-ΔCH and βII-spectrin-ΔPH did not show periodic distributions (Fig. 2h–k). We also observed similar results when these GFP-tagged βII-spectrin mutants were expressed in wild-type neurons instead of in the βII-spectrin knockout neurons (Supplementary Fig. 9): these truncated proteins did not exhibit periodic distributions, and overexpression of these truncated proteins also did not significantly disrupt the endogenous MPS structure. Together, these results suggest that both the actin-binding and membrane-binding functions of βII-spectrin are required for the MPS formation and/or maintenance.

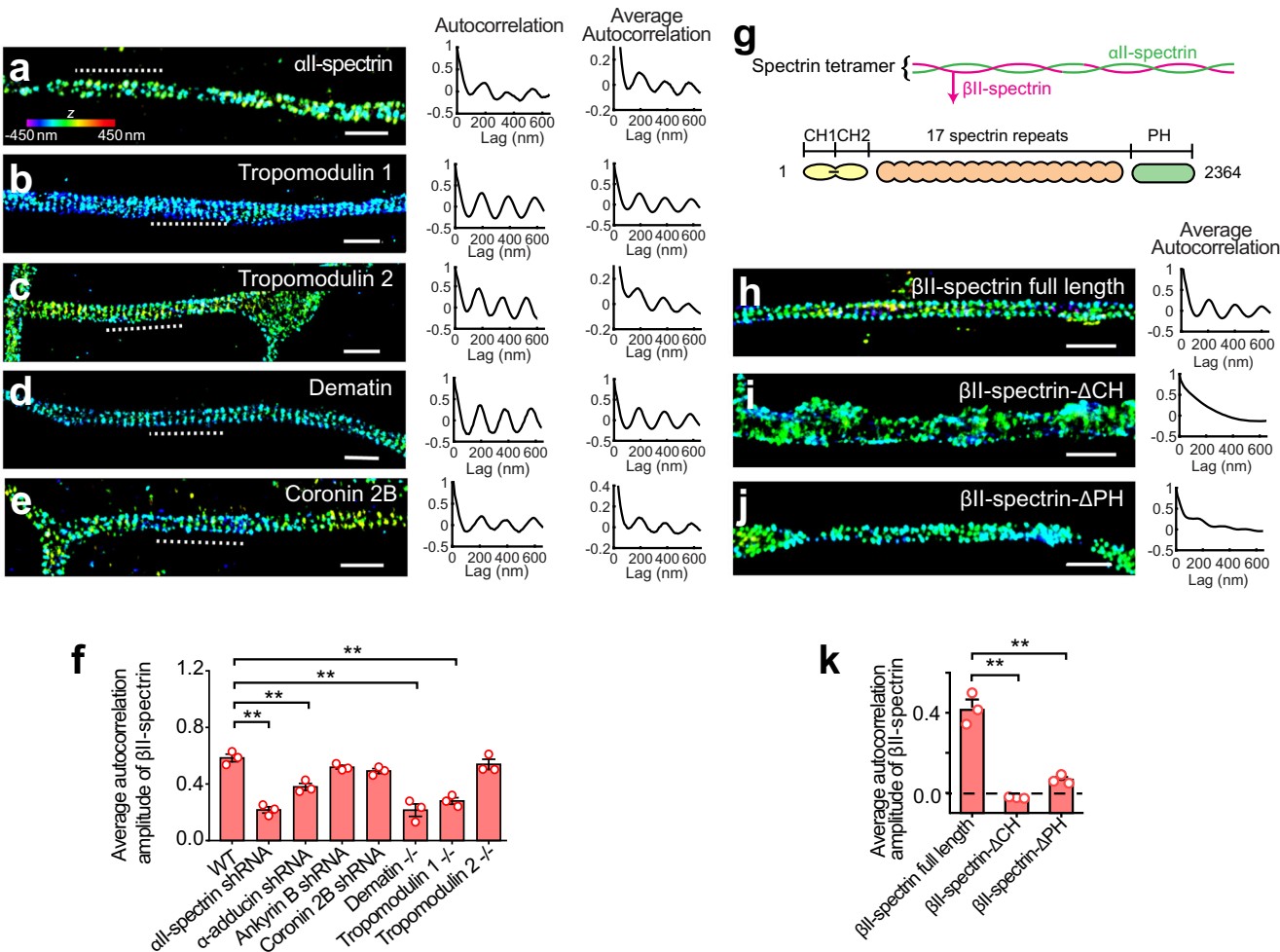

**Fig. 2 Super-resolution imaging of actin-binding MPS components. a–e** Left: 3D STORM images of αII-spectrin (**a**), tropomodulin 1 (**b**), tropomodulin 2 (**c**), dematin (**d**), and coronin 2B (**e**) in axonal regions of cultured hippocampal neurons. Middle: One-dimensional (1D) autocorrelation of the imaged molecules for the axon region indicated by the dashed line in the left panels. Signals were projected to the longitudinal axis of the axon segment, and 1D autocorrelation was calculated using the projected signals. Right: Average 1D autocorrelation of the imaged molecules over 20–80 randomly selected axon regions. Scale bars: 1 µm. Colored scale bar indicates the z-coordinates. **f** Average 1D autocorrelation amplitudes of the βII-spectrin distribution, indicating the degree of periodicity of the MPS, of untreated neurons, neurons transfected with adenoviruses expressing αII-spectrin shRNA, α-adducin shRNA, ankyrin B shRNA, or coronin 2B shRNA, and neurons cultured from tropomodulin 1, tropomodulin 2 or dematin knockout mice. Data are mean ± s.e.m. ($n = 3$ biological replicates for each condition; 40–80 axonal regions were examined for each condition). \*\*$p < 0.005$ (two-sided unpaired Student's t-test); *p-value*s (from left to right): $6.0 \times 10^{-4}$, $4.9 \times 10^{-3}$, $4.0 \times 10^{-3}$ and $1.1 \times 10^{-3}$. **g** Domain organization of βII-spectrin. **h–j** Left: 3D STORM image of axon regions of βII-spectrin knockout neurons transfected with plasmid expressing GFP-tagged full length βII-spectrin (**h**), βII-spectrin-ΔCH mutant (**i**) or βII-spectrin-ΔPH mutant (**j**). The proteins were visualized by immunostaining with anti-GFP antibody. Scale bars: 1 µm. Right: average 1D autocorrelation of the imaged proteins. **k** Average 1D auto-correlation amplitudes for the distributions of GFP-tagged full length βII-spectrin and two GFP-tagged βII-spectrin truncation mutants, as described in (**h–j**), calculated from GFP-positive axon segments. Data are mean ± s.e.m. ($n = 3$ biological replicates for each condition; 30–70 axonal regions were examined for each condition). \*\*$p < 0.005$ (two-sided unpaired Student's t-test); p-values (from left to right): $1.3 \times 10^{-3}$ and $1.4 \times 10^{-4}$. STORM images in **a–e** and **h–j** are representative examples from three independent experiments with similar results. Source data are provided in the Source Data file.

**Interactions of the MPS with the non-muscle myosin II (NMII) and role of the MPS in axon diameter regulation.** Another family of actin-binding proteins identified by our co-IP and mass spectrometry experiments as potential candidate MPS-interacting proteins were NMII motor proteins (Supplementary Data 1). NMII proteins were expressed throughout the axons of cultured neurons (Supplementary Fig. 10a). Phosphorylated myosin regulatory light chain 2, an activator of contractile NMII, has been shown to colocalize with actin rings in the AIS of cultured neurons[28,29], suggesting that the NMII bipolar filaments could bind to actin rings in the MPS and potentially generate tension to control the morphology and contractility of neurites. The NMII bipolar filaments could interact with MPS in two potential

binding modes: the motor domains at the two ends of the 300-nm long NMII bipolar filaments[30–34] may bind to the same actin ring or connect different actin rings in the MPS, providing the possibilities to control the radial and longitudinal contractility of axons, respectively.

To probe how NMII binds to the MPS, we used two-color STORM to examine the center and ends of the NMII bipolar filaments (marked by the C- and N-terminus of NMII heavy chain, respectively) with respect to the MPS (Fig. 3a). If each NMII filament binds within the same actin ring in the MPS (the first binding mode), both C- and N-terminal labels of NMII would be colocalized with the actin rings and hence form alternating patterns with the spectrin tetramer centers, marked by

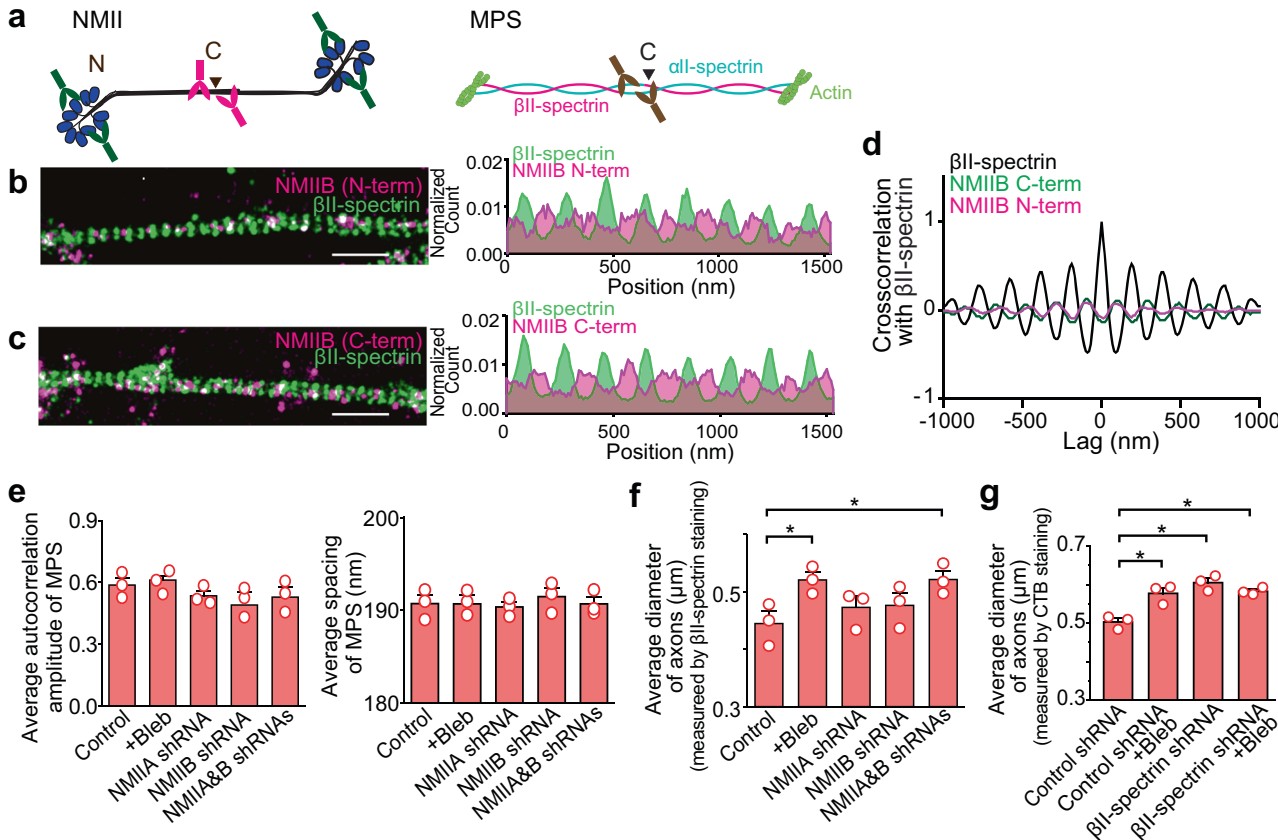

**Fig. 3 Non-muscle myosin II (NMII) filaments bind to the MPS and play a role in radial contractility of axons. a** Antibody-binding epitopes at the C- or N-terminus of a NMII bipolar filament (left), and the antibody-binding epitope at the C-terminus of βII-spectrin located near the center of the spectrin tetramer. **b** Left: Two-color STORM image of βII-spectrin (C-terminus, green) and NMIIB (N-terminus, magenta) in axons of cultured hippocampal neurons. Right: Average 1D distributions of βII-spectrin (C-terminus, green) and NMIIB (N-terminus, magenta) signals projected to the longitudinal axon axis from many axon segments. Scale bar: 1 μm. **c** Similar to (**b**) but for the C-terminus of NMIIB. **d** Average 1D cross-correlation between the distributions of βII-spectrin (C-terminus) and NMIIB (N-terminus) (magenta) and between the distributions of βII-spectrin (C-terminus) and NMIIB (C-terminus) (green) along the axons of cultured neurons, derived from 20–50 axon segments. The average 1D auto-correlation of βII-spectrin distribution (black) is shown as a reference. **e** Average 1D autocorrelation amplitudes of βII-spectrin distribution (left) and average periodic spacing of the MPS (right), for axons of untreated neurons, Blebbistatin (Bleb)-treated neurons, and neurons transfected with adenoviruses expressing shRNAs against NMIIA, NMIIB, or both NMIIA and NMIIB heavy chains. **f** Average diameter of axons (measured by βII spectrin immunostainining) for untreated neurons, Bleb-treated neurons, and neurons treated with shRNAs against NMIIA, NMIIB, or both NMIIA and NMIIB heavy chains. *$p < 0.05$ (two-sided unpaired student's t-test); *p-value*s (from left to right): $4.7 \times 10^{-2}$ and $4.7 \times 10^{-2}$. **g** Average diameter of axons (measured by CTB staining) for neurons treated with control (scramble) shRNA, neurons treated with control shRNA and Bleb, neurons treated with βII-spectrin shRNA, and neurons treated with βII-spectrin shRNA and Bleb. *$p < 0.05$ (two-sided unpaired student's t-test); *p-value*s (from left to right): $3.2 \times 10^{-2}$, $1.0 \times 10^{-2}$ and $3.3 \times 10^{-2}$. Data are mean ± s.e.m ($n = 3$ biological replicates; 50–130 axonal regions per condition). STORM images in **b**–**c** are representative examples from three independent experiments with similar results. Source data are provided in the Source Data file.

the C-terminus of βII-spectrin (Fig. 3a). For the second binding mode, if each NMII filament connects adjacent actin rings, only the N-terminus of NMII would be colocalized with the actin rings but the C-terminus of NMII would be colocalized with the spectrin tetramer centers. Our STORM images showed that both C- and N-termini of NMIIB, and the N-terminus of NMIIA (a good antibody against C-terminus of NMIIA is lacking), preferentially localized midway between the periodic stripes formed by the spectrin tetramer centers (Fig. 3b–d and Supplementary Fig. 10b). As this midway position is where the actin rings are situated, these results suggest that among the two binding modes, the first mode where the NMII filaments bind within the same actin rings is the preferred one. The sparse appearance of NMII filaments in the STORM images could be because the labeling efficiency of the NMII antibodies is relatively low, or the number of NMII filaments bound to each actin ring is

small. It is also worth noting that there is a non-zero baseline in the average 1D distribution of NMII (N-terminus or C terminus) along the axon axis (Fig. 3b, c and Supplementary Fig. 10b). This non-zero baseline could represent 1) NMII filaments not bound to actin or NMII filaments bound to non-MPS actin filaments, 2) nonspecific binding of the NMII antibodies, and/or 3) a fraction of NMII filaments adopting the second binding mode where the NMII bipolar filaments connect adjacent actin rings in the MPS. In addition, in the second binding mode, if the NMII filaments could bridge every other actin ring spaced ~380 nm apart, both C- and N-terminal labels of NMII would also appear to be colocalized with the actin rings. Although we cannot rule out this possibility, we consider this scenario less likely because it would require NMII filaments connecting every other actin ring to be stretched to ~380 nm long whereas the mean length of NMII bipolar filaments was measured to be 300 nm with a standard

deviation of 20 nm by super-resolution fluorescence imaging[34], or 290–320 nm with a standard deviation of 20–30 nm by electron microscopy[30,33].

Next, we examined whether the axon diameter is changed upon inhibition of the NMII activity by treatment with Blebbistatin (Bleb), a drug that inhibits NMII motor function and prevents tension-generation by bipolar filaments[35]. Bleb treatment (Supplementary Fig. 10c) did not significantly disrupt the MPS structure or its 190 nm spacing (Fig. 3e and Supplementary Fig. 10d) but increased the average axon diameter by 20–30% (Fig. 3f). In addition to Bleb treatment, knockdown of both NMIIA and NMIIB (Supplementary Fig. 10e) also led to a similar increase in the average axon diameter without changing the average spacing between adjacent actin rings (Fig. 3e, f and Supplementary Fig. 10d). These results are consistent with the model that NMII filaments bind to actin filaments within individual rings, forming actomyosin complexes that radially contract the actin rings and control the axon diameter.

To further assess whether the effect of NMII inhibition is through the interaction of NMII with the MPS or with other actin structures, we further examined whether Bleb treatment could also increase the average axon diameter of neurons in which the MPS had been disrupted by βII-spectrin knockdown (Supplementary Fig. 3a). In the experiments described above, we used βII-spectrin immunostaining to estimate the axon diameter. Here, to allow measurements of the axon diameter in βII-spectrin knockdown neurons, we used cholera toxin B (CTB) to label the axonal membrane, and as expected, the axon diameters measured by CTB were slightly larger than those measured by βII-spectrin, which is beneath the axonal membrane. We found that βII-spectrin knockdown led to an 20–30% increase in the average axon diameter (Fig. 3g and Supplementary Fig. 10f) without significantly changing the expression levels of NMIIA and NMIIB in axons (Supplementary Fig. 10e). The degree of axon diameter enlargement induced by βII-spectrin knockdown was similar to that induced by Bleb treatment or by NMIIA and NMIIB knockdown of WT neurons. Furthermore, Bleb treatment did not further increase the average axon diameter of βII-spectrin knockdown neurons (Fig. 3g), suggesting a lack of MPS-independent mechanisms for the NMII-mediated axon radial contraction.

**Transmembrane proteins and membrane-associated signaling proteins interacting with the MPS.** Among the 480 candidate MPS-interacting proteins commonly identified in the three co-IP experiments from cultured mouse hippocampal neurons using all three bait proteins, αII-spectrin, βII-spectrin, and α-adducin (Supplementary Data 1), only 19 were transmembrane proteins and some transmembrane proteins known to be associated with the MPS, such as sodium channels[1], were missing. The low abundance of transmembrane proteins identified was likely because transmembrane proteins are difficult to immunoprecipitate due to their low solubility in the co-IP lysis buffer. To obtain a more inclusive list of candidate MPS-interacting transmembrane proteins, we pooled together all transmembrane proteins identified from the three pulldown experiments, rather than only selecting the common proteins detected in all three pulldown experiments. The resulting pooled list contained 95 transmembrane proteins (Supplementary Data 7), among which several have been previously shown to associate with the MPS and exhibit periodic distributions at the AIS or the node of Ranvier, including the sodium channel Na_v, and two cell adhesion molecules (Neurofascin and NrCAM)[1,6,11,12]. GO term enrichment analyses of these transmembrane proteins showed enrichment for ion transmembrane transporter activity, transmembrane receptor protein

kinase activity, neurotransmitter receptor activity, channel activity, axon guidance receptor activity, transmembrane signaling receptor activity, protein binding involved in cell-cell adhesion, etc. (Supplementary Data 8). When we performed the similar proteomic analyses for the three co-IP experiments from the whole mouse brain lysates using αII-spectrin, βII-spectrin, or α-adducin as the bait, 176 transmembrane proteins were identified, and the enriched GO terms had a substantial overlap with those determined for cultured mouse hippocampal neurons (Supplementary Fig. 11 and Supplementary Data 8). As a cautionary note, although pooling together all the transmembrane proteins from the three co-IP experiments using different baits could generate a more inclusive list of candidate proteins that directly or indirectly interact with the MPS, it could also increase the number of falsely identified MPS-interacting transmembrane proteins as well as the number of the corresponding enriched GO terms.

To test the interactions of some of these transmembrane proteins with the MPS, we applied STORM imaging to visualize several representative candidate proteins with different known cellular functions. In the ion channel category, we observed that K_v1.2 and K_v1.3, exhibited periodic distributions with 190-nm spacing along axons (Fig. 4a). In the cell adhesion molecule category, we observed that Neurofascin, NrCAM, L1CAM, NCAM1 and CHL1 exhibited periodic distributions with 190-nm spacing and different degrees of periodicity as indicated by the autocorrelation amplitude (Fig. 4b). Compared to Neurofascin, NrCAM and CHL1, the relatively low degree of periodicity for L1CAM and NCAM1 could be due to the challenges in labeling as described earlier (Supplementary Fig. 4). Unlike Neurofascin, NrCAM and CHL1, which were labeled using antibodies against the endogenous proteins, L1CAM and NCAM1 were labeled using expression of GFP-fusion proteins through low-titer lentiviral transfection followed by immunolabeling using anti-GFP antibody because we could not identify antibodies against L1CAM and NCAM1 with sufficient specificity. Alternatively, the relatively low degree of periodicity could also be due to the existence of a considerable fraction of the examined molecules that were not localized to the MPS.

Since GO term analyses revealed that some candidate MPS-interacting proteins are involved in cell signaling, we also imaged several non-transmembrane, but membrane-associated signaling proteins from the candidate MPS-interacting protein list (Supplementary Data 1). Interestingly, we found that calcium/calmodulin-dependent protein kinase IIβ (CAMK IIβ) exhibited periodic distributions with 190-nm spacing along axons (Fig. 4c). The heterotrimeric G protein subunits also exhibited a tendency, albeit weak, to adopt the 190-nm periodicity (Fig. 4c). In addition, a membrane-associated signaling protein important for axon outgrowth, brain acid soluble protein 1 (BASP1)[36], as well as a transmembrane axon-growth-related protein, glycoprotein M6A[37], also exhibited periodic distributions along axons (Fig. 4c). These results suggest that the MPS may play a role in various signaling pathways in neurons.

**Role of the MPS in neurite-neurite interactions mediated by cell adhesion molecules.** In the brain, cell adhesion molecules play essential roles in regulating neuronal migration and neurite outgrowth, and in establishing and maintaining various types of neuron-neuron contacts, such as axon-axon fasciculation (i.e., bundling), axon-dendrite fasciculation, as well as synaptic interactions[38–42]. Because our experiments revealed MPS interactions with several cell adhesion molecules, it raises the possibility that the MPS serves as a structural platform to anchor these molecules and facilitate adhesion-molecule-mediated contacts between neurites.

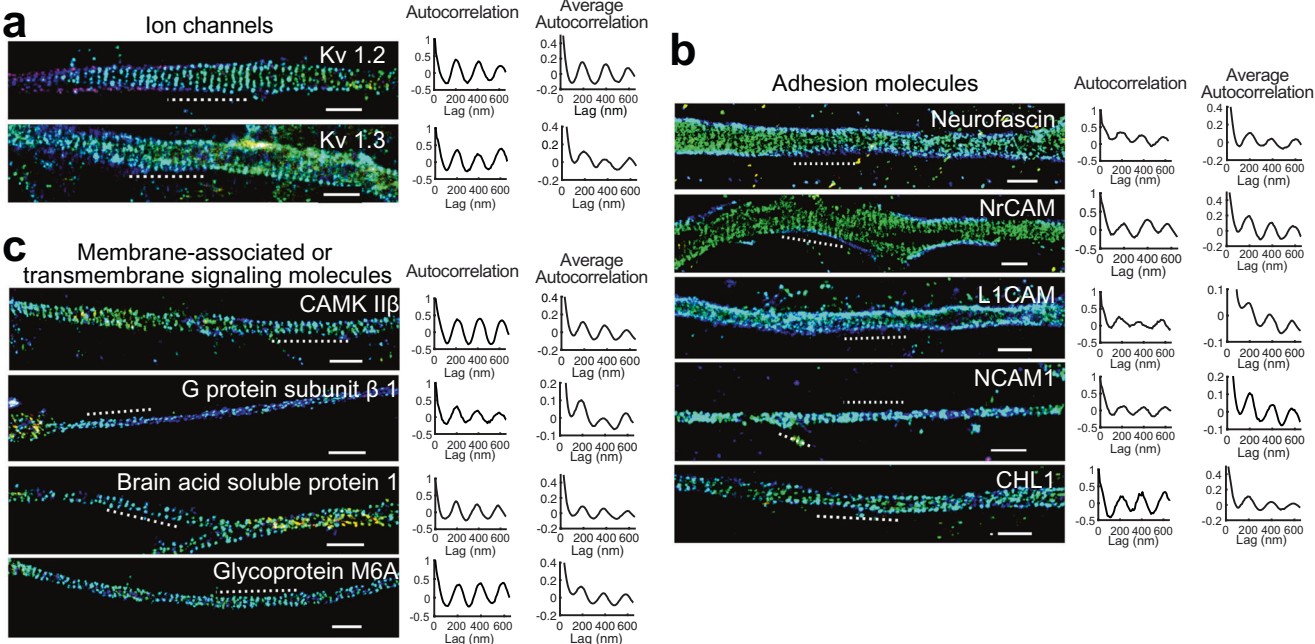

**Fig. 4 STORM imaging of transmembrane proteins or membrane-bound signaling proteins associated with the MPS. a** Left: 3D STORM images of two potassium channel subunits, $K_v$ 1.2 and $K_v$ 1.3, in axons. Middle: 1D autocorrelation of the imaged proteins for the axon region indicated by the dashed line in the left panels. Right: Average 1D autocorrelation of the imaged proteins over 20–90 randomly chosen axon regions. **b** Same as (**a**) but for five cell adhesion molecules, including neurofascin, NrCAM, L1CAM, NCAM1 and CHL1. **c** Same as (**a**) but for three membrane-associated (non-transmembrane) signaling molecules, including calcium/calmodulin-dependent protein kinase type IIβ (CAMK IIβ), heterotrimeric G protein β-subunit 1, and brain acid soluble protein 1, as well as a transmembrane signaling molecule, glycoprotein M6A. STORM images in **a–c** are representative examples from three independent experiments with similar results. Scale bars: 1 μm. Source data are provided as a Source Data file.

If the MPS-associated adhesion molecules indeed can bring neurites into contact, we expect that the MPS structures in the abutting neurites would be aligned in phase with each other by the contacting pairs of adhesion molecules, which bind to defined sites in the MPS. Indeed, we observed that the periodic distributions of βII-spectrin in the abutting axon-axon and axon-dendrite pairs tended to be in phase (Supplementary Fig. 12), consistent with the previous observation of in-phase alignment of the MPS of two contacting axons[12].

To test whether the MPS structure is indeed involved in the formation of axon-axon and axon-dendrite interactions, we examined how axon-axon and axon-dendrite fasciculations were affected by βII-spectrin knockdown, which is known to disrupt the MPS[4,7], or by knocking down ankyrin B, a protein that can interact with both the MPS[1] and cell adhesion molecules[2] and hence may serve as an adaptor to anchor these adhesion molecules to the MPS.

Notably, axon-axon bundling was substantially reduced by either βII-spectrin or ankyrin B knockdown (Fig. 5a). In order to quantify the axon-bundling effect, we used an automated algorithm to determine the average width of the axon-positive areas in the images (including both axon bundles and single axons) and found that this width parameter was significantly decreased in both βII-spectrin knockdown neurons and ankyrin B knockdown neurons, as compared to control neurons treated by scrambled shRNA (Fig. 5a). The average width values observed in these βII-spectrin knockdown and ankyrin B knockdown samples reduced to values that were close to those observed from single, unbundled axons (Fig. 5a), indicating a substantial disruption of axon-axon bundling when the MPS was disrupted or when the adaptor protein between cell adhesion molecules and the MPS was depleted. To examine the knockdown effects on axon-dendrite interactions, we quantified two properties: 1) axon-

dendrite bundling, defined by the average fraction of dendrite segments that had adhering axons; and 2) the density of synaptic contacts along dendrites. We imaged synaptic contacts using immunofluorescence against presynaptic scaffolding protein bassoon and postsynaptic density protein homer1 and counted puncta that were positive of both bassoon and homer1 as synapses. We found that βII-spectrin knockdown and ankyrin B knockdown caused a significant decrease both in the degrees of axon-dendrite bundling and in the synapse density along dendrites (Fig. 5b, c). These data suggest that the MPS plays a role in axon-axon and axon-dendrite interactions.

Next, we examined the effect of knocking down MPS-associated cell adhesion molecules on axon-axon and axon-dendrite interactions. Among the adhesion molecules that are associated with the MPS (Fig. 4b), we examined two that are known to be enriched in axons, L1CAM and CHL1[43–45] and one that is enriched in both axons and dendrites, NCAM1[43], by knocking down these proteins (Supplementary Fig. 13). We did not study the effect of knocking down Neurofascin and NrCAM because they are only enriched in the AIS region[6,11], a relatively short segment in axons. Knockdown of L1CAM or NCAM1 decreased the axon-axon and axon-dendrite bundling, whereas knockdown of CHL1 did not show any significant effect (Fig. 5a, b). The density of synapses was reduced only in the NCAM1 knockdown neurons, but not in the L1CAM or CHL1 knockdown neurons (Fig. 5c), consistent with the results from previous studies[44–46]. When all three adhesion molecules were knocked down, axon-axon bundling, axon-dendrite bundling, and synapse density were all reduced (Fig. 5a–c). These data suggest that L1CAM and NCAM1 are involved in axon-axon and axon-dendrite fasciculations, that NCAM1 is additionally involved in synapse formation, whereas CHL1 is not required for any of these three types of neurite interactions.

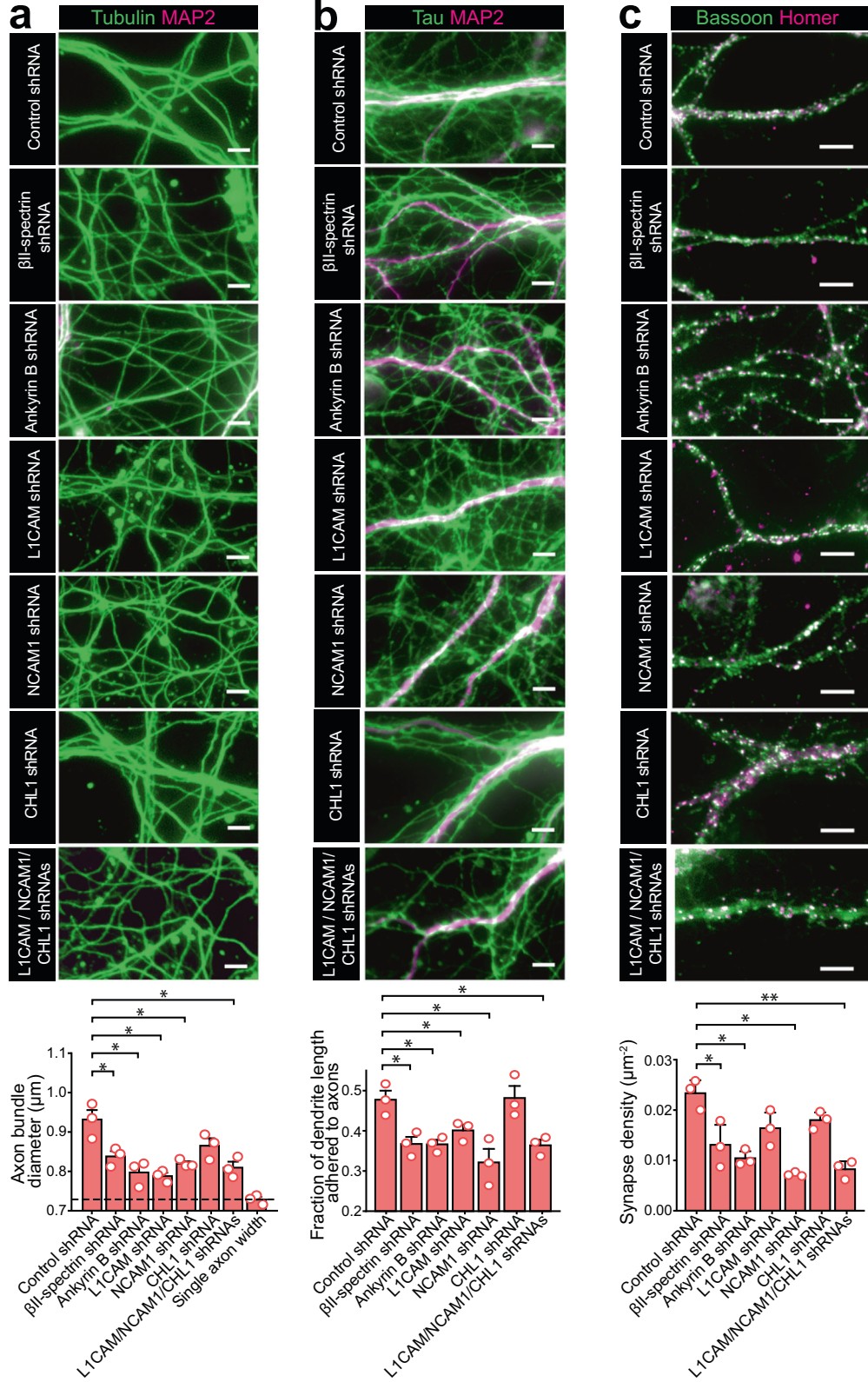

Our βII spectrin knockdown and ankyrin B knockdown experiments suggested that the MPS plays a role in axon-axon and axon-dendrite interactions, and our L1CAM and NCAM1 knockdown experiments, as well as previous studies[40,42,45,46], implicated these cell adhesion molecules in axon-axon and axon-dendrite contacts. An interesting question arises as to whether the role of the MPS in axon-axon and axon-dendrite contacts is simply to maintain proper expression levels of these cell adhesion molecules at the neurite surface. We thus examined whether the observed reduction in axon-axon and axon-dendrite interactions in βII-spectrin knockdown or ankyrin B knockdown neurons could be simply explained by a reduction in the cell-surface expression levels of the cell adhesion molecules by measuring the amounts of L1CAM, NCAM1 and CHL1 at the surface of

**Fig. 5 The MPS plays a role in axon-axon interactions, axon-dendrite interactions and synapse formation. a** Top panels: Conventional fluorescence images of tubulin (green) and dendrite marker MAP2 (magenta) for neurons transfected with adenoviruses expressing scrambled (control) shRNA, βII-spectrin shRNA, ankyrin B shRNA, L1CAM shRNA, NCAM1 shRNA, CHL1 shRNA, or shRNAs against all three adhesion molecules (L1CAM, NCAM1 and CHL1). The tubulin-positive neurites lacking MAP2 signal are axons. Bottom panel: Average diameter of axon bundles (or single axons) quantified for the conditions described in the top panels. The last bar and dashed line represent the average single-axon diameter obtained from images of sparsely cultured neurons. * indicates $p < 0.05$ (two-sided unpaired student's t-test); *p-value*s (from left to right): $3.9 \times 10^{-2}$, $1.3 \times 10^{-2}$, $1.7 \times 10^{-2}$, $3.7 \times 10^{-2}$, $1.8 \times 10^{-2}$ and $9.3 \times 10^{-3}$. **b** Top: Conventional fluorescence images of cultured neurons immunostained for axon marker Tau (green) and dendrite marker MAP2 (magenta) under the conditions described in (**a**). Bottom: Average fraction of the total length of dendrites that are bundled with axons, quantified for the conditions indicated in the top panel. *$p < 0.05$ (two-sided unpaired student's t-test); *p-value*s (from left to right): $1.7 \times 10^{-2}$, $1.8 \times 10^{-2}$, $4.9 \times 10^{-2}$, $2.4 \times 10^{-2}$ and $1.5 \times 10^{-2}$. **c** Top: Conventional fluorescence images of cultured neurons immunostained for the presynaptic marker Bassoon (green) and postsynaptic marker Homer1 (magenta) under the conditions described in (**a**). Bottom: Average synapse density per unit area on dendrites, quantified for the conditions as described in the top panel. Only puncta showing both presynaptic and postsynaptic marker signals were counted as synapses. *$p < 0.05$ and **$p < 0.005$ (two-sided unpaired student's t-test);*p-value*s (from left to right): $3.8 \times 10^{-2}$, $6.4 \times 10^{-3}$, $9.6 \times 10^{-3}$ and $3.3 \times 10^{-3}$. Scale bars: 5 μm. Data are mean ± s.e.m ($n = 3$ biological replicates; 15–25 imaged regions per condition). Images in **a–c** are representative examples from three independent experiments with similar results. Source data are provided in the Source Data file.

neurites. βII-spectrin or ankyrin B knockdown led to only a small reduction in the cell-surface expression levels of L1CAM, and the cell-surface expression levels of NCAM1 and CHL1 even increased moderately (Supplementary Fig. 14). It is thus unlikely that the effect on the cell-surface expression levels of cell adhesion molecules could fully account for the substantial reduction in axon-axon bundling, axon-dendrite bundling, and synapse density observed in βII-spectrin knockdown or ankyrin B knockdown cells. Hence, our data suggest that spatial organization of the cell adhesion molecules by the MPS likely help establish and/or maintain efficient molecular contacts at the interface of abutting neurites.

## Discussion

In this study, we combined co-IP and mass spectrometry to identify candidate proteins that interact directly or indirectly with the neuronal MPS at the proteomic scale. We identified hundreds of potential candidate MPS-interacting proteins, including cytoskeletal proteins and their binding proteins, transmembrane proteins, and membrane-associated signaling molecules. Using super-resolution microscopy, we imaged ~20 of these potential candidate MPS-interacting proteins, including structural components of the MPS that can bind to actin filaments, non-muscle myosin II (NMII) motor proteins, cell-adhesion molecules, ion channels, and other signal transduction related proteins, and showed that these proteins indeed form periodic distributions along axons.

Based on these results, and prior knowledge of the neuronal MPS[47–49] and the erythrocyte membrane skeleton[2,3], we propose a more comprehensive picture of the molecular architecture of the MPS and its interacting proteins in neurons (Fig. 6). The neuronal MPS is comprised of actin rings periodically spaced by spectrin tetramers along the neurites. The actin ring consists of actin filaments, actin-capping proteins, including adducin and tropomodulins, and other actin-binding proteins, including dematin and coronins. In addition, many transmembrane proteins, such as ion channels, cell adhesion molecules, receptors, and glycoproteins, may interact with the neuronal MPS. These transmembrane proteins may be associated with either the actin rings through adaptor proteins such as dematin or associated with the spectrin tetramers through adaptor proteins such as ankyrins, and both association modes have been previously observed to connect some transmembrane proteins to the erythrocyte membrane skeleton[2,3].

Compared to the erythrocyte membrane skeleton, the neuronal MPS exhibits several notable distinctions. First, in addition to a few shared structural components (tropomodulin 1, α-adducin,

and dematin) used by both structures, the neuronal MPS contains distinct protein homologs and unique proteins. For example, αII-spectrin, βII-spectrin, βIII-spectrin, βIV-spectrin, ankyrin B, and ankyrin G are protein homologs specifically observed in the neuronal MPS, whereas the counterparts in the erythrocyte membrane skeleton are αI-spectrin, βI-spectrin and ankyrin R, respectively. Moreover, many proteins associated directly or indirectly with the neuronal MPS have not been found on the erythrocyte membrane skeleton, such as structural proteins (e.g., coronins), transmembrane proteins (e.g., sodium and potassium ion channels, GPCRs, RTKs), and membrane-associated signaling molecules (e.g., CAMK IIβ). Ultrastructure wise, the erythrocyte membrane skeleton adopts a two-dimensional polygonal lattice structure[2,3], whereas the neuronal membrane skeleton adopts a one-dimensional periodic structure in axons and portions of dendrites and a two-dimensional polygonal lattice structure in the neuronal cell body and portions of dendrites[1,4,6,7]. We observed actin-capping proteins in the MPS structure in neurites, including both adducin and tropomodulin. It has been shown previously that adducin binds to the fast-growing-end of actin filaments 25-fold stronger than to the sides of actin filaments[50], and tropomodulin, in conjunction with tropomyosin, is a slow-growing end capping protein and cannot bind along the sides of actin filaments[51]. Our observations thus suggest the possibility that the actin filaments in the periodic actin rings in the neuronal MPS are capped at both ends. However, it remains an open question whether the lengths of these actin filaments in neuronal MPS are short 12–17mers as observed for the erythrocyte membrane skeleton[3], or whether their lengths could be longer and more heterogeneous[29].

Moreover, we observed that tension-generating non-muscle myosin II (NMII) motors are associated with the neuronal MPS. Our observation that both C- and N-termini of the NMII preferentially colocalized with the actin rings favors the model that NMII bipolar filaments bind within individual actin rings, although our data do not exclude the possibility that a smaller fraction of NMII filaments connect the adjacent actin rings in the MPS (Fig. 6). Therefore, NMII filaments can potentially form actomyosin complexes with the actin filaments in individual actin rings and exert contractile forces radially to control the diameter of the neurite. Indeed, we observed that acute inhibition of NMII activity by Blebbistatin treatment, depletion of both NMIIA and NMIIB heavy chains, or depletion of βII-spectrin all resulted in 20–30% increase of average axon diameter. Consistent with our observations, two parallel studies showed that NMII filaments interact with the MPS and that inhibition of NMII increases the average axon diameter[52,53]. This radial contractility of axons could be functionally important because the action potential

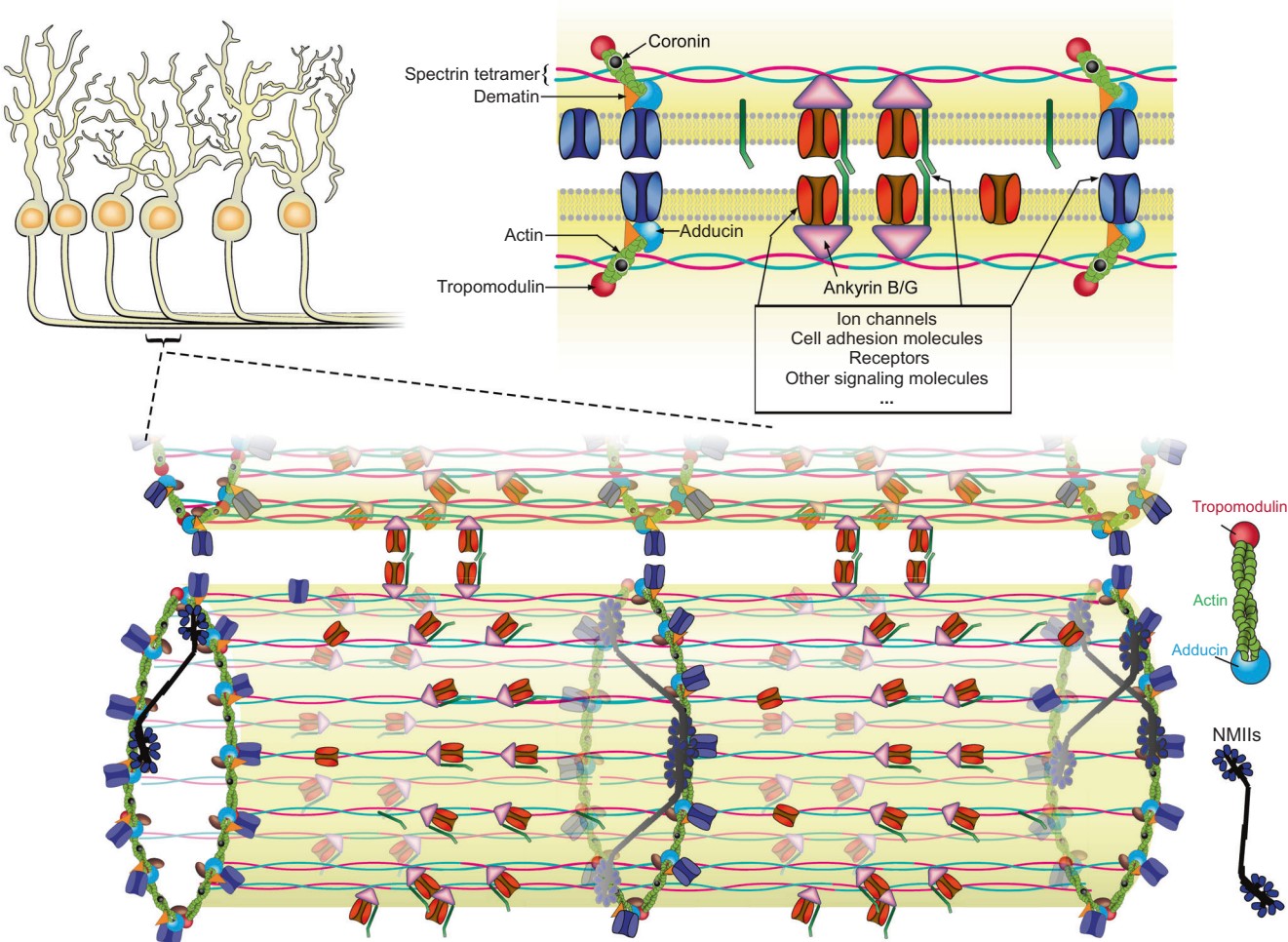

**Fig. 6 Molecular architecture model of the neuronal MPS in the neurite-neurite fasciculations.** In neurites, actin filaments form ring-like structures that are connected by spectrin tetramers, and actin filaments in these rings are likely capped by adducin and tropomodulin at their fast- and slow-growing ends, respectively. The length and structure of the actin filaments within the actin rings remains an open question and a recent electron microscopy study suggested the possibility that actin rings in the AIS are made of long, intertwined actin filaments[29]. Dematin and coronin bind to the actin filaments, and ankyrin B and ankyrin G bind to the site near the center of each spectrin tetramer. The NMII bipolar filaments bind preferentially to actin filaments within the same actin rings and may regulate the diameter of neurites by exerting radial contractile forces. It is possible that a smaller fraction of NMII bipolar filaments may connect adjacent actin rings. Membrane proteins, such as ion channels, cell adhesion molecules, receptors and signaling molecules, are associated either with the actin filaments through adaptor proteins such as dematin, or with the center positions of spectrin tetramers through adaptor proteins such as ankyrin. Recruitment of cell adhesion molecules to periodic sites on the MPS and interactions of cell adhesion molecules between two abutting neurites bring their respective MPS structures in phase, which could in turn bring additional MPS-bound cell adhesion molecules from the two neurites into proximity, enhancing neurite-neurite interactions. Moreover, the MPS could serve as a structural platform that organizes transmembrane proteins and membrane-associated signaling molecules, potentially facilitating a variety of signaling pathways in neurons.

conduction velocity along axons can change substantially upon a 10–20% axonal diameter change[54,55]. Indeed, inhibiting NMII activity increases axonal signal propagation velocity[53]. In addition, this NMII-mediated axon contractility has also been shown to affect cargo transport in axons[52].

Finally, we showed that the MPS plays a role in the establishment or maintenance of axon-axon and axon-dendrite interactions. MPS organizes cell adhesion molecules, such as L1CAM and NCAM1, into periodic distributions along neurites, and knockdown of L1CAM or NCAM1 led to a substantial reduction in axon-axon and axon-dendrite bundling. Importantly, disruption of the MPS by βII-spectrin knockdown or depletion of ankyrin B, the adaptor protein which connects cell adhesion molecules to the MPS, also caused a substantial reduction in axon-axon and axon-dendrite bundling, as well as in synaptic contacts between axons and dendrites, even though the

cell-surface expression levels of these cell-adhesion molecules did not decrease substantially under these conditions. We propose the following potential mechanisms for the MPS-dependent axon-axon and axon-dendrite interactions. First, recruitment of adhesion molecules to periodic sites on the MPS may substantially enhance the interactions of adhesion molecules between two abutting neurites, because contacts between some MPS-bound adhesion molecules from two abutting neurites may lead to in-phase alignment of the MPS structures contained in the two neurites, which may in turn bring additional MPS-bound adhesion molecules from the two neurites into proximity. In addition, it has been shown that downstream signaling pathways activated by adhesion molecules are required for axon outgrowth, guidance, and fasciculation[41,56,57]. It is thus also possible that MPS facilitates the neurite-neurite interactions by providing a structural platform to recruit relevant signaling molecules and adhesion

molecules to common sites to form signaling complexes and enable signaling relevant to neurite-neurite interactions. In support of this notion, we have recently shown that the MPS can function as a platform to recruit NCAM1 and related signaling molecules to enable NCAM1-induced ERK signaling[13]. Our mass spectrometry and super-resolution imaging results shown here further demonstrated that many other signaling molecules are associated with the MPS, suggesting the possibility that the MPS could serve as a structural platform to facilitate other signaling pathways involved in the establishment of neurite-neurite interactions, as well as in other neuronal functions.

## Methods

**Primary culture of mouse hippocampal neurons.** All experimental procedures were performed in accordance with the Guide for the Care and Use of Laboratory Animals of the National Institutes of Health. The protocols were approved by the Institutional Animal Care and Use Committee (IACUC) of Harvard University (protocol #10–16–3), Scripps Research Institute (protocol #08–0087), and Tufts University (protocol #B2021–159).

Primary cultures of hippocampal neurons were prepared as previously described[10]. The mouse lines used in this study include CFW mice (Strain code 024, Charles River Laboratories), a $\beta II$-$Spec^{flox/flox}$ mouse line[27], a Nestin-Cre mouse line (Stock number 003771, The Jackson Laboratory), a whole body Tmod1 knockout mouse line that expresses a $Tmod1$ transgene only in the heart[58], a whole body $Tmod2$ knockout mouse line[59], and a whole body $Dmtn$ knockout mouse line[60]. Mice were housed at an ambient temperature of 19–23 °C with a humidity of 55% (±10%) and 12 h dark/light cycle. To make hippocampal neuronal cultures, hippocampi were isolated from mouse embryos at embryonic day 18 (for CFW, $\beta II$-Specflox/flox, and Nestin-Cre mice) or at postnatal day 0 (for Tmod1 knockout, Tmod2 knockout, and Dmtn Knockout mice). For imaging experiments, 4–6 embryos or newborn pups were used per condition; for co-IP experiments, 10–15 embryos were used per condition. Mouse pups were not sexed, and we expect approximately equal amounts of males and females. Isolated hippocampi were digested with 0.25% trypsin-EDTA (1x) (Sigma, T4549) at 37 °C for 15 min. The digested tissues were washed in Hanks' Balanced Salt Solution (HBSS) (Thermo Fisher Scientific, 14175079) for three times, and then transferred to culture medium consisting of Neurobasal medium (Thermo Fisher Scientific, 21103049) supplemented with 37.5 mM NaCl, 2% (vol/vol) B27 supplement (Thermo Fisher Scientific, 17504044) and 1% (vol/vol) Glutamax (Thermo Fisher Scientific, 35050-061). The tissues were gently triturated in the culture medium until there were no chunks of tissue left. Dissociated cells were then counted and plated onto poly-D-lysine-coated 18 mm coverslips or 10 cm petri dishes. Cultures were maintained in the culture medium in a humidified atmosphere of 5% $CO_2$ at 37 °C. The neurons were fed with one-half medium volume change every five days. At DIV 2 (day in vitro), AraC (2 μM) was added to the culture medium to prevent glial overgrowth for all the neuronal cultures used for mass spectrometry analyses. For proteomic analyses on adult mouse brains, female CFW mice at the age of 8–12 weeks were used.

**U2OS and HEK293T cell culture.** Human bone osteosarcoma epithelial cells (U2OS cells) were obtained from ATCC. U2OS cells were plated onto 8-well Lab-Tek glass bottom chambers (Thermo Fisher Scientific, 155409) and maintained in complete growth Dulbecco's modified Eagle's medium (DMEM), supplemented with 10% (vol/vol) fetal bovine serum (FBS), 0.1% (vol/vol) penicillin–streptomycin. HEK293T cells were plated onto 10 cm Petri dishes and maintained in DMEM, supplemented with 10% (vol/vol) fetal bovine serum (FBS), 0.1% (vol/vol) penicillin–streptomycin

**Antibodies.** The following primary antibodies were used in this study: guinea pig anti-MAP2 antibody 1:500 dilution for immunofluorescence (IF) (Synaptic Systems, 188004), rabbit anti-MAP2 antibody 1:500 for IF (Synaptic Systems, 188002), mouse anti-αII spectrin antibody 1:400 for IF (Biolegend, 803201, Clone D8B7), mouse anti-αII spectrin antibody 1:200 for IF (Encor Biotechnology, MCA-3D7, Clone 3D7), rabbit anti-αII spectrin antibody 1:200 for IF (Encor Biotechnology, RPCA-aII-Spec), mouse anti-αII spectrin antibody 1:200 for IF (EMD Millipore, MAB1622, Clone AA6), mouse anti-βII spectrin antibody 1:200 for IF (BD Biosciences, 612563, Clone 42), mouse anti-dematin antibody 1:50 for IF (Santa Cruz Biotechnology, sc-135881, Clone 18), rabbit anti-coronin 2B antibody 1:200 for IF (Novus Biologicals, NBP 1-85567), mouse anti-tubulin antibody 1:100 for IF (Santa Cruz Biotechnology, sc-5286, Clone B7), rabbit anti-Tau antibody 1:500 for IF (Synaptic Systems, 314002), mouse anti-$K_v$1.2 channel antibody 1:200 for IF (Neuromab, 75-008, Clone K14/16), rabbit anti-neurofascin antibody 1:200 for IF (Neuromab, 75–172, Clone A12/18), rabbit anti-NrCAM 1:200 for IF (Abcam, ab24344), goat anti-CHL1 antibody 1:200 for IF (R&D systems, AF2147), rabbit anti-NCAM1 antibody 1:200 for IF (EMD Millipore, AB5032), mouse anti-ankyrin G antibody 1:100 for IF (Santa Cruz Biotechnology, sc-12719, Clone 463), mouse anti-bassoon antibody 1:400 for IF (Enzo, ADI-VAM-PS003-F, Clone SAP7F407),

rabbit anti-homer antibody 1:500 for IF (Synaptic Systems, 160003), rabbit anti-L1CAM antibody 1:500 for Western blot (WB) (ABclonal, A8555), rat anti-L1CAM antibody 1:200 for IF (R&D Systems, MAB5674, Clone 555), rabbit anti-NMIIB (Myh10) (N-terminus) antibody 1:200 for IF (GeneTex, GTX133378), rabbit anti-NMIIA (Myh9) (N-terminus) antibody 1:200 for IF (GeneTex, GTX101751), rabbit anti-NMIIB (Myh10) (C-terminus) antibody 1:200 for IF (Biolegend, 909901), rabbit anti-Glutamate Receptor 2 & 3 antibody 1:200 for IF (EMD Millipore, AB1506), rabbit anti-GFP antibody 1:400 for IF (Thermo Fisher Scientific, A11122). rabbit anti-β-actin antibody 1:1000 for WB (Proteintech, 20536-1-AP).

The following secondary antibodies were used in this study: CF680-conjugated donkey anti-mouse IgG antibody 1:400 for IF (Biotium, 20819), Alexa-647-conjugated donkey anti-mouse IgG antibody 1:800 for IF (Jackson ImmunoResearch, 715-605-151), Alexa-647-conjugated donkey anti-rabbit IgG antibody 1:800 for IF (Jackson ImmunoResearch, 711-605-152), Alexa-647-conjugated donkey anti-goat IgG antibody 1:800 for IF (Jackson ImmunoResearch 705-605-147), Alexa-647-conjugated donkey anti-rat IgG antibody 1:800 for IF (Jackson ImmunoResearch, 712-605-153), Cy3-conjugated donkey anti-rabbit IgG antibody 1:500 for IF (Jackson ImmunoResearch, 711-165-152), Cy3-conjugated donkey anti-guinea pig IgG antibody 1:800 for IF (Jackson ImmunoResearch, 706-165-148), Cy3-conjugated donkey anti-mouse IgG antibody 1:800 for IF (Jackson ImmunoResearch, 711-165-151). Alexa-488-conjugated donkey anti-guinea pig IgG antibody 1:800 for IF (Jackson ImmunoResearch, 706-545-148).

**Co-immunoprecipitation of MPS-interacting proteins for mass spectrometry analysis.** The magnetic protein-G-coated beads (Thermo Fisher Scientific, 10007D) were incubated with the irrelevant (control) antibody or antibody against βII-spectrin, αII-spectrin, or α-adducin at 4 °C overnight, followed by 3x washes using the washing buffer (PBS (pH 7.4) containing 0.02% (vol/vol) Tween-20). For whole mouse brain co-immunoprecipitation experiments, adult mouse brains were isolated and dounce-homogenized in the co-immunoprecipitation lysis buffer containing 25 mM Tris-HCl (pH 7.4), 150 mM NaCl, 1 mM EDTA, 5 mM EGTA, 1% NP-40, 5% glycerol, 1x protease inhibitor cocktail (Thermo Fisher Scientific, 87785) on ice. For co-immunoprecipitation experiments from cultured mouse hippocampal neurons, DIV 20 neurons were lysed on ice using the co-immunoprecipitation lysis buffer. After incubating for 15 min, cell lysates were centrifuged at 13,000 g for 5 min at 4 °C. The supernatant was collected and then incubated with the antibody-coated magnetic beads for 3 h at 4 °C. The magnetic beads with the co-immunoprecipitated products were washed 5 times using the washing buffer. Finally, the co-immunoprecipitated proteins were eluted using Laemmli sample buffer (Bio-Rad, #161-0737) for 10 min at 90 °C. The eluted protein mixtures from the whole adult mouse brains or cultured neurons (DIV 20) were then separated using SDS-PAGE, and each gel lane was cut into gel pieces and subjected to in-gel tryptic digestion as previously described[61,62].

**Sample preparation for quantitative mass spectrometry.** Adenoviruses expressing βII-spectrin shRNA or scrambled (control) shRNA were added to the cultured mouse hippocampal neurons at DIV 2. At DIV 20, cultured mouse hippocampal neurons treated with adenoviruses expressing either βII-spectrin shRNA or scrambled (control) shRNA were lysed on ice using the RIPA buffer (Thermo Fisher Scientific, 89900) complemented with 1x protease and phosphatase inhibitor cocktail (Thermo Fisher Scientific, 78440). Cell lysates were transferred to microcentrifuge tubes and centrifuged at 13,000 g for 5 min at 4 °C, and the supernatants were collected for quantitative mass-spectrometry analyses. Protein mixtures (three replicates for each condition) were digested and labeled by tandem mass tags (TMT) 6plex reagent (Thermo Fisher Scientific, 90061), according to the manufacturer's protocol.

**Protein identification using mass spectrometry.** Experiments were performed on a hybrid ion trap-Orbitrap mass spectrometer (Orbitrap Velos, Thermo Fisher Scientific) for all analytical runs. HPLC samples (Waters) were injected into the trap column (75 μm column ID, 5 cm packed with 5 μm beads on 200 Å pore size, from Michrom Bioresources, Inc.) and washed for 15 min, and were then eluted to the analytical column with a gradient from 2 to 32% of 0.1% formic acid over 90 min. The instrument was set up to run the TOP 20 method for MS/MS in ion trap. The raw mass spectrometry data were analyzed by Proteome Discoverer 2.4 (Thermo Fisher Scientific), and searches were performed against mouse (*Mus musculus*) Uniprot database and known common lab contaminants. 1% false discovery rate (FDR) was held for both protein and peptide levels for all output reported data.

To remove the proteins that are non-specifically pulled down by the antibody coated beads from the identified protein list, we performed co-IP experiments using the irrelevant (control) IgG antibody that is not directed to any known antigen. We then adapted the distributed Normalized Spectral Abundance Factor (dNSAF) previously reported as a label free quantitative measure of protein abundance based on the spectral counts which are corrected for peptides shared by multiple proteins[63]. For each co-IP experiment using αII-spectrin, βII-spectrin, or α-adducin as the bait, we calculated the fold-changes of dNSAF values in the capture experiment using the proper bait protein vs the negative control using

irrelevant IgG and removed the proteins with the dNSAF fold-changes that were not greater than 1 (See Supplementary Data 1 for the dNSAF values and the fold-changes).

**Plasmids construction and transfection of βII spectrin truncation mutants**. βII-spectrin-ΔCH-GFP and βII-spectrin-ΔPH-GFP plasmids were modified from FUGW-GFP plasmid (Addgene, 14883). In details, for βII-spectrin-ΔCH-GFP construct, 1–303 amino acids truncated βII-spectrin was amplified from wide-type βII-spectrin and inserted into FUGW-GFP. A GGGGS peptide linker was inserted between truncated βII-spectrin and GFP. Likewise, for βII-spectrin-ΔPH-GFP, 2169–2364 amino acids truncated βII spectrin was amplified from wide-type βII-spectrin and inserted into FUGW-GFP with a GGGGS peptide linker. Plasmids were transfected into cultured neurons at DIV 7–9 using a calcium phosphate transfection kit (Thermo Fisher, K2780-01). Experiments were performed 2 or 3 days post transfection.

**Expression of GFP fusion proteins in neurons**. Plasmids used in this study were purchased from GeneCopoeia or OriGene, and cloned into lentiviral expression vector FUGW (Addgene, 14883), or pLVX (Clontech, 632155), through Gibson assembly reaction (New England Biolabs, E2611S) according to the manufacturer's protocol. *Kcna3*, *Gnb1* and *Camk2b* were tagged at their N-terminus with the GFP gene and cloned into pLVX vector. *Add1*, *Tmod1*, *Tmod2*, *Actn1*, *L1cam*, *Ncam1*, *Gpm6a*, *Basp1* were tagged at their C-terminus with the GFP gene and cloned into FUGW vector.

Lentiviruses were then produced by co-transfecting HEK293T cells with 6 μg lentiviral expression vector (FUGW or pLVX), 4.5 μg Δ8.9 vector[64] (gift from Prof. David Baltimore, California Institute of Technology), and 3 μg VSVG packaging vector (Addgene, 8454) in a 100 mm dish using a calcium phosphate transfection kit (CalPhos Mammalian Transfection kit, Clontech, 631312). Two days post-transfection, the supernatant was harvested, centrifuged at 3000 rpm for 10 min, and then concentrated with Lenti-X concentrator (Clontech, 631231), before being snap frozen in liquid nitrogen. The lentivirus expressing the desired GFP-tagged protein was added to the neuronal cultures at DIV 5–8. The cultured neurons were then fixed for imaging between DIV 14–21.

**shRNA knockdown and drug treatments**. The sense sequences of the βII-spectrin shRNA are 5′-GCATGTCACGATGTTACAA-3′ and 5′-GGATGAAAT-GAAGGTGCTA-3′.

The sense sequence of ankyrin B is 5′-GACAAGCAGAAGTTGTCAA-3′.
The sense sequence of α-adducin is 5′- GTGACTGCATCCAGTTTGG-3′.
The sense sequence of αII-spectrin is 5′- AGCATGATGTTCAAACACT -3′.
The sense sequence of coronin 2B is 5′- CCATCACCAAGAATGTACAT -3′.
The sense sequences of NMIIA heavy chain shRNA and NMIIB heavy chain shRNA are 5′-TACCCTTTGAGAATCTGATAC-3′ and 5′-CTTCCAATTTACT CTGAGAA-3′, respectively.

The sense sequences of L1CAM shRNA, NCAM1 shRNA and CHL1 shRNA are 5′-TGCTAGCCAATGCCTACATTT-3′, 5′-CGTTGGAGAGTCCAAATTC TT-3′ and 5′-GCAGAAGATCAGGGTGTTT-3′, respectively.

The adenovirus expressing a scramble shRNA sequence was used as a control (Vector BioLabs, 1122). For knocking down the specific gene, adenoviruses expressing the above target sequences were added to the neuronal cultures at DIV 5 (for αII-spectrin shRNA treatment) and DIV 2–3 (for the rest of the shRNA treatments). The knockdown efficiencies were confirmed by either immunofluorescence or western blot analysis.

To perform the co-IP experiments from whole adult mouse brain under the condition where the MPS was disrupted, F-actin disrupting drugs (20 μM LatA and 40 μM CytoD) were added to the whole brain lysate immediately before incubating the lysate with the beads coated with the βII-spectrin antibody, and the F-actin disrupting drugs remained in the lysate during the entire period of this incubation step. To perform the co-IP experiments from cultured mouse hippocampal neurons (DIV 20) under the condition where the MPS was disrupted, 20 μM LatA and 40 μM CytoD were added to the neuronal culture medium for 2 h before the cultured hippocampal neurons were lysed, and 20 μM LatA and 40 μM CytoD were also added to the culture neuron lysate as described for the brain tissue lysate.

For Blebbistatin (Bleb) treatment, cultured mouse hippocampal neurons (DIV 14) or the U2OS cells were treated with 100 μM Bleb for 2 h at 37 °C.

**Cell fixation and immunostaining for fluorescence imaging**. For STORM imaging, cultured mouse hippocampal neurons untreated or treated with adenoviruses expressing various shRNAs or lentivirus expressing various GFP-tagged proteins were fixed between DIV 14–21 using 4% (w/v) paraformaldehyde (PFA) in phosphate buffered saline (PBS) for 30 min at room temperature (RT), washed three times in PBS, and permeabilized with 0.15% (vol/vol) Triton X-100 in PBS for 10 min. Neurons were then blocked in blocking buffer containing 3% (wt/vol) bovine serum albumin (BSA) in PBS for 1 h, and subsequently stained with primary antibodies in blocking buffer overnight at 4 °C. Neurons were washed three times with PBS and stained with fluorophore conjugated secondary antibodies in blocking buffer for 1 h at RT. The samples were post-fixed with 4% PFA for 20 min.

For conventional fluorescence imaging, U2OS cells were fixed and immunostained using the same protocol described above.

αII-spectrin, dematin and coronin 2B were immunolabeled with their corresponding antibodies against the SH3 domain of human αII-spectrin, the N-terminal amino acids 68–190 of human dematin, and the C-terminal amino acids 364–467 of human coronin 2B, respectively. Tropomodulin 1 and tropomodulin 2 were labeled using moderate expression of GFP-tagged proteins, with GFP fused to the C-terminus of the target protein, through low-titer lentiviral transfection, which were in turn immunolabeled using anti-GFP antibody. GFP-tagged tropomodulin 1 was expressed in tropomodulin 1 knockout neurons; GFP-tagged tropomodulin 2 was expressed in WT neurons. K$_v$ 1.2, neurofascin, NrCAM, CHL1 were immunolabeled with their corresponding antibodies raised against the C-terminal amino acids 428–499 of K$_v$ 1.2, the extracellular domain of rat neurofascin, the extracellular region of NrCAM, and recombinant mouse CHL1, respectively. K$_v$ 1.3, BASP1, glycoprotein M6A, NCAM1, CAMK IIβ, and G protein β-subunit 1 were labeled using moderate expression of GFP-tagged proteins, with GFP fused to the C-terminus of K$_v$ 1.3, BASP1, glycoprotein M6A, and NCAM1, or the N-terminus of CAMK IIβ and G protein β-subunit 1, through low-titer lentiviral transfection of WT neurons, which were in turn immunolabeled with anti-GFP antibody.

To measure the average axon diameter, cultured mouse hippocampal neurons (untreated, treated with adenoviruses expressing various shRNAs, or treated with Bleb) were fixed at DIV 14 and stained with either anti-βII-spectrin antibody (BD Biosciences, 612563) or Cholera Toxin Subunit B (Thermo Fisher Scientific, C34778).

To quantify the neurite-neurite interactions using the conventional fluorescence imaging, cultured mouse hippocampal neurons treated with adenoviruses expressing various shRNAs were fixed at DIV 21 and immunostained with the dendrite marker MAP2, the axon marker Tau, tubulin, the presynaptic marker Bassoon, and the postsynaptic marker Homer1 using the same protocol described above.

For immunostaining of the cell-surface NCAM1, CHL1 and L1CAM, neurons at DIV 21 were fixed using 4% PFA in PBS and blocked in blocking buffer for 1 h. The primary antibody detecting the extracellular domain of NCAM1, CHL1 or L1CAM was then added to label the cell-surface NCAM1, CHL1 and L1CAM, respectively, and neurons were incubated with the primary antibody for 1 h at RT. Neurons were washed three times with PBS and stained with fluorophore conjugated secondary antibodies in blocking buffer for 1 h at RT. The samples were post-fixed with 4% PFA for 20 min.

**STORM imaging**. The STORM setup was based on a Nikon Eclipse-Ti inverted microscope or Olympus IX71. 405 nm (OBIS 405–50 C; Coherent) and 647 nm (F-04306-113; MPB Communications) lasers were introduced into the sample through the back port of the microscope. A translation stage allowed the laser beams to be shifted towards the edge of the objective so that the emerging light reached the sample at incidence angles slightly smaller than the critical angle of the glass-water interface, thus illuminating only the fluorophores within a few micrometers of the coverslip surface. T660LPXR (Chroma) was used as the dichroic mirror and an ET705/72 M band-pass filter (Chroma) was used as the emission filter. For one-color three-dimensional (3D) STORM imaging, a cylindrical lens was inserted between the microscope side port and the EMCCD camera (Andor iXon, DU-897E-CSO-#BV, Andor Technology) so that images of single molecules were elongated in x and y for molecules on the proximal and distal sides of the focal plane (relative to the objective), respectively, and the ellipticity of the single-molecule images were used to determine the z position of the molecules[22]. For two-color two-dimensional STORM imaging, a multichannel imaging system (Photometrics, QV2) was inserted between microscope body and the EMCCD camera. Using the QV2 system, the emission light was split by T685LPXR (Chroma) into the two channels for the two-color STORM imaging acquisition. As a result, signals from CF680 and Alexa 647 were split into two color-channels, and the fluorophores were identified by measuring the intensity ratio of each single molecule in two channels[65,66].

The sample was imaged in PBS buffer containing 100 mM cysteamine (Sigma), 5% glucose (Sigma), 0.8 mg/mL glucose oxidase (Sigma), and 40 μg/mL catalase (Roche Applied Science). During imaging, continuous illumination of 647 nm laser (~2 kW/cm²) was used to excite fluorescence from Alexa 647 or CF680 molecules and switch them off. Continuous illumination of the 405-nm laser was used to reactivate the fluorophores to the emitting state. The intensity of the activation laser (0–1 W/cm²) was adjusted during image acquisition so that at any given instant, only a small fraction of the fluorophores in the sample were in the emitting state, such that their images are optically resolvable, and their positions can be determined from their images.

A typical STORM image was generated from a sequence of about 25,000~40,000 image frames at a frame rate of 60 Hz. The recorded single-color 3D STORM movie was analyzed according to previously described methods using Insight3[22]. Briefly, the centroid positions and ellipticities of the single-molecule images provided lateral and axial positions of each activated fluorescent molecule, respectively[22]. Super-resolution images were reconstructed from the molecular coordinates by depicting each location as a 2D Gaussian peak. For two-color STORM analysis, each channel was analyzed using the same method as the single-color STORM analysis. Two channels were then aligned, and the intensity ratio of

each aligned fluorophore between two channels was determined and used to assign the fluorophore color[65,66].

**GO term enrichment analyses.** The GO term enrichment and clustering analyses were performed using the Database for Annotation, Visualization, and Integrated Discovery (DAVID 6.8, https://david.ncifcrf.gov/)[67]. The official gene symbols of the candidate MPS-interacting proteins were submitted to DAVID, and the enriched GO terms with $p$-value ≤ 0.05 were considered as significant. For GO term enrichment and clustering analyses of biological process (BP) category, GO BP terms at level 3 (GO_BP_3) were used. For GO term enrichment analysis of molecular function (MF) category, GO MF terms at level 4 (GO_MF_4) were used.

**Autocorrelation analysis and cross-correlation analysis.** For 1D autocorrelation analysis of each molecule, the signals in the STORM images were projected to the longitudinal axis of the axon segments, and autocorrelation functions were calculated from these 1D projected signals. For the average autocorrelation function, the autocorrelation curves from many randomly selected axon segments (~2 μm in length; the number of segments are specified in figure captions) were averaged for each condition. The average 1D autocorrelation amplitude was defined as the difference between the first peak (at ~190 nm) and the average of the two first valleys (at ~95 nm and ~285 nm) of the average 1D autocorrelation curve, which provides a quantitative measure of the degree of periodicity of the measured distribution in the neurites[4,10]. The average period (or spacing) of the MPS structure was defined as the first peak position of 1D average autocorrelation function. For 1D cross-correlation analysis between two molecules (e.g., NMIIA/B and βII spectrin), the signals in the two color-channels of the STORM images were each projected to the longitudinal axis of the axon segments, and a cross-correlation function was calculated from these projected signals from the two color channels for each axon segment, and then averaged over many randomly selected axonal segments (~2 μm in length; the number of segments are specified in figure captions) for each condition. To obtain the average 1D distributions of βII-spectrin and NMIIB signals projected to the longitudinal axon axis, the axonal regions were aligned based on the phase of the βII-spectrin distribution for each analyzed region and the average 1D distributions were then calculated for both βII-spectrin and NMIIB.

**Axon-axon and axon-dendrite bundling analyses.** For axon-axon bundling, the morphology of immunostained axons in the field of view of conventional fluorescence images were determined using Otsu's threshold, and the average width of axon-axon bundles were calculated using the ImageJ plugin DiameterJ, a previously validated nanofiber diameter characterization tool to find the central axis and average diameter of tubular structures in an image[68]. For axon-dendrite interaction, the degree of axon-dendrite bundling was calculated as the fraction of the dendritic length that shows overlapping axon and dendrite signals. The synapse density was obtained by calculating the number of colocalized (within two camera pixels, i.e., ~266 nm) pre-synaptic and post-synaptic markers divided by total dendrite area.

**Reporting summary.** Further information on research design is available in the Nature Research Reporting Summary linked to this article.

## Data availability

All data supporting the findings of this study are provided within the paper and its supplementary information. The fasta file of the mouse proteome (Uniprot *Mus musculus* proteome UP000000589) was downloaded from Uniprot. The mass spectrometry proteomics data have been deposited to the ProteomeXchange Consortium via the PRIDE partner repository[69] with the data set identifier PXD030886. Source data of all data presented in graphs within the figures are provided with this paper. Uncropped gel images are shown in Supplementary Fig. 15.

## Code availability

Custom codes for STORM image acquisition are available at https://github.com/ZhuangLab/storm-control and Zenodo (https://doi.org/10.5281/zenodo.3264857.)[70]. Custom codes for the two-color STORM data analysis, autocorrelation, and cross-correlation analysis are available at https://github.com/boranhan/MPS and Zenodo (https://doi.org/10.5281/zenodo.6513278.)[71].

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

## Acknowledgements

We thank Jiang He for help with plasmid construction, Ke Xu for providing the software for ratiometric two-color STORM analysis, Bogdan Budnik and Renee Robinson for help with mass-spectrometry analyses, and Toshihiko Hanada for the generation of conditional whole body Dmtn null mice. This work is supported in part by the National Institutes of Health. R.Z. was an HHMI Fellow of the Life Sciences Research Foundation. X.Z. is a Howard Hughes Medical Institute investigator.

## Author contributions

R.Z., B.H., and X.Z. designed the experiments with input from A.H.C. and V.M.F.; R.Z., B.H., R.N, Y.L, E.H, and C.X. performed the experiments. B.H. and R.Z. performed data analysis. R.Z., B.H., and X.Z. interpreted the data and wrote the manuscript with input from A.H.C., V.M.F., and the other authors.

## Competing interests

The authors declare no competing interests.
