## [Peer Review File · Nature Communications]

Proteomic and functional analyses of the periodic membrane skeleton in neuronsREVIEWER COMMENTS

Reviewer #1 (Remarks to the Author):

In the paper “Proteomic and functional analyses of the periodic membrane skeleton in neurons” from R. Zhou et al. the authors use co-immunoprecipitation followed by mass spectrometry to identify membrane-associated periodic skeleton (MPS) binding partners in the brain and the primary neuron culture. The identified targets were validated using the STORM microscopy and their functional impact on the radial contractility of axons, the localization of transmembrane proteins, and the axon-axon / axon-dendrites interaction were described.

Key conclusions are:

- I. Identification of tropomodulin 1, tropomodulin 2, dematin and coronin 2B as new structural components of neuronal MPS. Among them, the loss of α -spectrin, α -adducin, tropomodulin 1, and dematin led to a significant disruption of the MPS, as measured by the degree of the periodicity of the β II-spectrin distribution in axons.
- II. MPS members can interact with non-muscle myosin (NMII) and regulate axon diameter.
- III. Due to its interaction with several TM proteins the MPS might play a role in the regulation of signaling from the plasma membrane.
- IV. The MPS plays a role in axon-axon and axon-dendrite interaction. β II-spectrin and Ankyrin B knockdown decrease the axon bundle diameter and the dendrite length adhered to the axon, causing a decrease in the synapse density.

In general, the manuscript is well-written and is of high interest to the cell biology/neuroscience community. I found the concept interesting. On the other hand, I missing the rationale behind several experiments and this has to be clarified before the manuscript can be published.

Major comments:

1. It is not clear why Fig. 1B contains a mixture of identified proteins from the brain and the primary culture? Is this to compare the binding partners in-vivo and in-vitro? This has to be explained. Also, please provide the MS results from all-spectrin and all-adductin IP from culture neurons as well. Please indicate how many biological replicates (for brain and culture) were measured for each co-immunoprecipitation experiment (β II-spectrin, α -spectrin and α -adducin). Also, please show the Q- (False discovery rate) Values for these binding partners. In fact, the GO molecular terms analysis in figure 4a illustrates that the transmembrane proteins pulled by the β II-spectrin differ substantially between the primary neurons and the brain.
2. Since all the candidates tested in this study show a periodic distribution, it would be good to see a negative control for STORM-based imaging, such a protein which are not associated with the MPS, and thus lack a periodic pattern of localization. This is really crucial since it will illustrate the specificity of tested candidates to the MPS in neurons.
3. The first paragraph states: “Magnetic beads were coated with the antibody that can specifically bind to a bait protein previously known to be an MPS structural component, such as β II-spectrin, α -spectrin and α -adducin, ...”. Later, however, the α -spectrin is chosen as one of the five candidates for validation. Please clarify the rationale behind this. If the authors question the role of α -spectrin in the MPS, why it was used in co-IP experiments?

4. The last paragraph states: “ β II-spectrin or ankyrin B knockdown only led to a small reduction in the cell-surface expression levels of L1CAM, but did not decrease the cell-surface expression levels of NCAM1 or CHL1 (Supplementary Fig. 8).” But the data in Fig. S8 show a significant increase in surface levels of NCAM1 and CHL1, or am I mistaken? In case the graph is correct, could the increase in the surface levels of NCAM1 and CHL1 be the cause of alterations in the neurite bundling and synapse density?

Minor:

1. Please describe the criteria, which were used to choose the validation targets (α II-spectrin, tropomodulin 1, tropomodulin 2, dematin and coronin 2B) from the list of 515 immunoprecipitated proteins?
2. I am missing the evaluation of β II-spectrin KD efficiency for analysis in Fig. 1d? KD efficiency is analyzed as a part of Fig. S5f, but this is not clear if these results also belong to Fig. 1.
3. To verify the targets, the authors could also perform the IP and MS analysis under the β II-spectrin KD conditions.
1. The MPS structure is disrupted by the knockdown of several MPS candidates validated by the average autocorrelation amplitude of β II-spectrin (Fig. 2f) leading to increased axon diameter. Is the diameter of the dendrites also increased or is this phenotype specific to axons? Do the neurons show an altered morphology / complexity?
2. In Fig. S1a & 1b please indicate the input amount of proteins that were given to the antibody-coupled beads from brain lysates and lysed cultured neurons.
3. I am missing the number (n) of axons analyzed per experiment in Fig. 4.
4. Please add an example illustrating KD experiments in Suppl. Figure 4a.
5. For Fig. S8 please indicate the p-values in b and c, as well as the experimental N.
6. Discussion part is too long and can be shortened by half.

Reviewer #2 (Remarks to the Author):

The manuscript shows evidence for proteome in neurons interacting directly or indirectly with proteins known to be main components of the MPS. To do so, mass spectrometry is conducted in protein pull-down obtained from 4 independent co-IP experiments: samples obtained from co-IP experiments against α 2-spectrin, β 2-spectrin or alpha adducin applied in mouse whole brain lysates and against β 2-spectrin applied in 20DIV hippocampal neuronal cultures. To narrow down MPS-interacting candidate proteins, authors consider those which are present in all 4 co-IP experiments, arriving to a list of 515 protein candidates.

The manuscript then evaluate some specific proteins by super-resolution microscopy to determine if these organize periodically in axons from hippocampal neurons in culture, in a way consistent with the MPS (190nm lags). These include known actin-interacting proteins and also signaling proteins.

The manuscript then evaluates some other candidates for their impact their knock-down has in the MPS structure, as seen by β 2-spectrin STORM imaging, providing evidence for proteins involved in MPS structure itself.

Finally, authors examine more specific protein families or groups for their biological function by loss-of-function approaches. Authors arrive to the conclusion that non-muscle myosin II (NMII)

function within a same actin “ring” to maintain axon diameter, a series of MPS-interacting transmembrane adhesion proteins function to allow a normal interaction of the axon to other axons or dendrites.

General concerns:

- The manuscript is weak in building upon a vast existing literature that can be used to enrich the interpretation of their own results: These include works describing the axonal proteome (Zappulo et al 2017.) and previous direct protein-protein interaction biochemical evidence. Zappulo A, Bruck D Van Den, Ciolli Mattioli C, Franke V, Imami K, McShane E, Moreno-Estelles M, Calviello L, Filipchuk A, Peguero-Sanchez E, Müller T, Woehler A, et al. 2017. RNA localization is a key determinant of neurite-enriched proteome. *Nat Commun* 8:583.

- The manuscript is weak in organizing the vast evidence obtained into a proposal of a way in organizing a model of the MPS. For instance, others have tried to do so with the existing evidence and suggested organizing MPS-related proteins into “structural components” (those than when knocked down disrupt the MPS), proteins that directly interacts with structural components (but not substantially altering the MPS if absent), and proteins that will evidence a periodic distribution but not directly associated with the MPS. It is nothing else than trying to agree on what IS the MPS and what is ASSOCIATED with it. This will avoid confusions when using the term MPS. These reviews tried to organize this information: Leterrier et al 2017 and Unsain et al 2018

Leterrier C, Dubey P, Roy S. The nano-architecture of the axonal cytoskeleton. *Nat Rev Neurosci.* 2017 Dec;18(12):713-726. doi: 10.1038/nrn.2017.129. Epub 2017 Nov 3. PMID: 29097785.

Unsain N, Stefani FD, Cáceres A. The Actin/Spectrin Membrane-Associated Periodic Skeleton in Neurons. *Front Synaptic Neurosci.* 2018 May 23;10:10. doi: 10.3389/fnsyn.2018.00010. PMID: 29875650; PMCID: PMC5974029.

- STORM imaging as a validation for protein-protein interactions is overstated.

Throughout the text, authors state that STORM imaging was used to validate interactions first obtained by the co-IP experiments. I do not agree with that. An experiment to validate interactions typically would look for interactions or close proximity by other means, like FRET, Bimolecular Fluorescence Complementation Analysis, Surface plasmon resonance, etc. Using super-resolution microscopy would be useful if co-detections are made and it is clearly calculated for each experiment what is the resolution attained, and then one can say proteins A and B are present in the same volume of approx. 30-60 nm xy. In summary, evidence presented by STORM imaging show that the candidate proteins show a periodic distribution consistent with the MPS, but is not a validation of interaction. Also, being a work mainly based on protein-protein interactions, it is striking that are virtually no co-immuno-fluorescence detection using STORM, and not at all imaging for filamentous actin, which is a central player in the MPS and the focus of figure 2.

I strongly suggest these validation claims are changed and that evidence from co-immunolabeling STORM with the newly discovered MPS-related proteins is included, to determine their position with respect to structural components of the MPS (namely filamentous actin and bli-spectrin).

Particulars

-Page 3, Paragraph 1. Author's lists proteins as MPS "interacting proteins", but for most of them interaction evidence was lacking, albeit have been shown to exhibit a periodic organization... but do not necessarily interact with the MPS. Again to do such a claim it is necessary that authors define more clearly what the MPS is.

Same comment can be done in Page 11. "...previously shown to bind to the MPS and exhibit periodic distributions...".

Again, the interpretation of previous work is overstated. In the literature, evidence of binding of these proteins to the MPS is mostly lacking. A protein can show MPS-induced periodic distribution without directly binding to its structural components. I believe that given the main goal of the present report, authors should be more specific about the evidence from the literature and explain in detail each of the possibilities.

-Page 4, 2nd paragraph, "Furthermore, treatment with actin disrupting drugs (LatA and CytoD), which are known to disrupt the MPS"... What was treated, the neurons in culture or the lysates, or the full-downs?, How? For how long? Dose? For the little information provided I interpret as being added to the protein lysates... By the way these drugs affect the dynamic turnover of actin filaments I don't see how these can even be effective in a cell-free system. Please clarify the details of the experimental approach and then its interpretation.

Also, for a completeness of the interpretation of the data, authors must mention that LatA and CytoD will disrupt the MPS BUT ALSO MANY other actin-related biological processes such as endo- and exocytosis, Golgi apparatus dynamics, vesicle transport to the vicinity of the membrane, among others.

-On the right side of supp fig 1 a and b, the names of certain proteins appear. The IgG can be seen by one of the lanes, but all the others are only estimates, since the stain cannot pick up individual proteins. Please clarify that or I would suggest to chance labels or identify individual proteins by immunolabeling. There are good arguments to pick up some visible bands as being actin and spectrin or IgG chains... but I don't agree that is true for alpha-adducin.

-Page 4, 3rd paragraph, spelling: "...tissues, i) using α -adducin as..." should say iii).

-"...of the MPS structures, for both 1D and 2D forms, in axons, dendrites and soma..." In the literature, no one uses the term MPS to refer to the topology of these components in the soma, referred by the authors as 2D. If that was the case, then erythrocytes would have an MPS. Also, the "abundance" of these 2D topologies in somas is very low. I suggest authors just acknowledge that they might be getting interactors that are not present in the MPS of axons and dendrites, since they have lysed the whole cell.

-In general, the results description should include some key experimental details for an easier appreciation of the results. For example, to appreciate the quantitative mass-spec upon beta spec knockdown, it is key to know the timing of the knockdown and subsequent analyses. Most interventions lack key information more at hand in the Results section. The Mat and Met section also lacks that information. I strongly suggest that for every shRNA experiment or OE or drugs treatment, authors identify WHEN it is treated (be it shRNA or drugs) and when samples are collected or fixed. This is very important for a correct interpretation of the data, especially to acknowledge that if interventions of cells in culture are performed early, they might be affecting

formation of the MPS, while when treatments are much later, they might be affecting maintenance of it.

-When comparing antibodies, it should be specified whether they target the same or different epitopes in the target protein, since that can be a source of disruption of the periodic visualization. This will be fair to acknowledge. Some proteins have long moving tails that even if well detected will not show a periodic structure, as is the case for the N- and C- terminus of Ankyrins.

-Figure 2. If actin binding partners are being tested by STORM, why aren't they compared directly to actin distribution? Or at least shown in the same set of experiments? Or compared that they are off-phase compared to beta spec? This is important since authors then build up a model (Fig 6) based mostly on previous knowledge, but not from a direct observation within the MPS. i.e. What is their evidence for tropomodulin or dematin to interact with the actin rings?

-Figure 2, Experiment with PH and CH deletions. To assess if b2-spec deletions really disturb the MPS, STORM imaging should be performed with at least one other structural component of the MPS, apart from b2-spec (actin, a2-spec, tropomodulin 1, dematin, etc). Is it the MPS itself disrupted or is it the way (or capability) of truncated b2-spec to be incorporated in the MPS that is disrupted?

Yours sincerely,
Nicolás Unsain, PhD

Reviewer #3 (Remarks to the Author):

The manuscript by Zhou and co-authors reports the membrane-associated periodic skeleton (MPS) proteome in neurons, and performs high-resolution microscopy and functional analysis to validate some of the new MPS components in neurons.

I have a major methodological concern on the proteomics data and how it is reported. Particularly with the experiments directed to identify MPS neuronal components (Fig. 1), which represent the foundations of the rest of the manuscript. Overall the detail on the MS methods used is insufficient (i.e. I could not find information on the number of replicates performed, are there biological or technical replicates?). Particularly concerning is how the authors dealt with proteins identified in the negative control IPs. Actually, it is unclear to this reviewer if they performed MS analysis on the negative control samples. I could not find in the manuscript or the supplementary tables the proteins identified in the negative control.

Analysing the negative control is key in order to define the true partners of betaII-spectrin, alphaII-spectrin and alpha-adducin. The authors must analyse the negative control (if they haven't) and report the proteins therein. Finally, they must explain how they subtract the data from the negative controls from the data in the problem IPs. Spectrins are very abundant

proteins and I wouldn't be surprised if there are traces of these proteins found in the negative control IP. In this case the authors should use a quantitative criterion to remove proteins identified in the negative control from those present in the problem IPs. If this has been done it should be properly reported. If it hasn't been done it should be done as this is key to remove false positives from problem IPs. The set of 515 MPS components would not be trust-worthy if the proteins in the negative control have not been removed from those in the problem IPs.

The authors present SDS-PAGE images of IP'ed samples, including the negative control. There it is clear that the negative control pulls-down far less proteins than problem IPs (Suppl Fig 1), indicating that the IPs are selective. Yet, a clear protein staining is visible in the negative control lanes, indicating that there are proteins spuriously IPed. This would be totally expected. It is a limitation of IPs and, in general, of affinity purification procedures. As mass spectrometry is very sensitive, many proteins will be identified in the negative control. Again, the authors should find a way to control for this.

From a formal stand point the results sections results very long (12 pages), including many elements of discussion. As a result the discussion is a bit redundant. Results could be written in a more succinct manner, leaving the discussion aspects for the final section of the manuscript.

Minor comment:

In the primary cultures, did the authors use drugs to prevent glial (or other mitotic cells growth?). This is quite standard in the field and it would be particularly important in this manuscript, as they aim at identifying the neuronal MPS proteome (not the glia MPS proteome).

Proteomics data should be made available through public repositories (i.e. PRIDE) and the corresponding link provided in the manuscript.

Response to reviewer comments

Reviewer #1 (Remarks to the Author):

In the paper “Proteomic and functional analyses of the periodic membrane skeleton in neurons” from R. Zhou et al. the authors use co-immunoprecipitation followed by mass spectrometry to identify membrane-associated periodic skeleton (MPS) binding partners in the brain and the primary neuron culture. The identified targets were validated using the STORM microscopy and their functional impact on the radial contractility of axons, the localization of transmembrane proteins, and the axon-axon / axon-dendrites interaction were described.

Key conclusions are:

- I. Identification of tropomodulin 1, tropomodulin 2, dematin and coronin 2B as new structural components of neuronal MPS. Among them, the loss of α -spectrin, α -adducin, tropomodulin 1, and dematin led to a significant disruption of the MPS, as measured by the degree of the periodicity of the β II-spectrin distribution in axons.
- II. MPS members can interact with non-muscle myosin (NMII) and regulate axon diameter.
- III. Due to its interaction with several TM proteins the MPS might play a role in the regulation of signaling from the plasma membrane.
- IV. The MPS plays a role in axon-axon and axon-dendrite interaction. β II-spectrin and Ankyrin B knockdown decrease the axon bundle diameter and the dendrite length adhered to the axon, causing a decrease in the synapse density.

In general, the manuscript is well-written and is of high interest to the cell biology/neuroscience community. I found the concept interesting. On the other hand, I missing the rationale behind several experiments and this has to be clarified before the manuscript can be published.

Response: We thank the reviewer for his/her enthusiasm about our manuscript and for providing constructive suggestions below, which we have addressed with additional experiments and analyses. These additional results further support our conclusions and strengthen our manuscript.

Major comments:

1. It is not clear why Fig. 1B contains a mixture of identified proteins from the brain and the primary culture? Is this to compare the binding partners in-vivo and in-vitro? This has to be explained. Also, please provide the MS results from all-spectrin and all-adductin IP from culture neurons as well. Please indicate how many biological replicates (for brain and culture) were measured for each co-immunoprecipitation experiment (β II-spectrin, α II-spectrin and α -adducin). Also, please show the Q- (False discovery rate) Values for these binding partners. In fact, the GO molecular terms analysis in figure 4a illustrates that the transmembrane proteins pulled by the β II-spectrin differ substantially between the primary neurons and the brain.

Response: We thank the reviewer for these suggestions. In the original manuscript, we mixed the co-immunoprecipitation (co-IP) results obtained from the brain and the primary culture and highlighted the proteins that were commonly identified in all co-IP experiments, with the thought that this may further reduce the number of potential false positives of the identified MPS structural components and interacting proteins. But as the reviewer pointed out, this could result in unnecessary confusions. To address this comment from the reviewer, we now revised the manuscript to analyze and report the results from primary neuronal culture and from whole brain lysates separately. In addition, we have performed new co-IP experiments for cultured neurons using α II-spectrin and α -adducin as the baits, such that our cultured neuron analyses also

include three baits (β II-spectrin, α II-spectrin and α -adducin), like what we did for brain tissues. In new Fig. 1b, we plot the Venn diagrams separately for cultured neurons and for whole-brain lysates, where 480 proteins were commonly identified using the three different bait proteins in the cultured-neuron co-IP experiments and 670 proteins were commonly identified using the three different bait proteins in the whole-brain co-IP experiments. The two lists of identified proteins showed substantial overlap: 321 of 480 candidate MPS-interacting proteins identified from cultured hippocampal neurons were also identified from whole brain lysates. The remainder of the 480 proteins identified from cultured hippocampal neurons but not from the whole brain lysates could be due to false positives or real biological differences (e.g. *in-vivo* versus *in-vitro*). More proteins were identified from the whole brain than from cultured neurons. Since the brain contains not only hippocampal neurons but also neurons from other brain regions as well as non-neuronal cells, it is not surprising that the number of proteins identified from the whole brain lysates is greater. We also performed GO term analysis for the two lists of proteins separately and observed substantial overlap between the GO terms enriched in these two lists of proteins (Fig. 1d, e). We note that the co-IP-based mass spec analysis provide a list of candidate MPS-interacting proteins, not all of which are necessarily associated with the MPS. Hence both lists of proteins could contain false positives. Additional experiments are needed to validate the direct or indirect association of these proteins with the MPS, as we pointed out in the manuscript (Page 4, last paragraph, Page 5, 1st paragraph, and Page 6, 2nd paragraph). We also performed validation for a subset of the identified candidate proteins, as shown in the manuscript (Figs. 2, 3, 4, and Supplementary Fig. 6).

As suggested by the reviewer, we now indicated how many biological replicates were measured (two biological replicates for each co-IP condition) in the Fig.1 caption, and we also included the Q-values (False discovery rate) and Sum PEP scores (The posterior error probability (PEP) is the probability that the observed peptide spectral count is incorrect) for all the identified proteins in the Supplementary Tables 1 and 7.

The reviewer mentioned that our original Fig. 4a illustrates that the transmembrane proteins identified as MPS-interacting proteins from cultured neurons are substantially different from those determined from the brain. This is not surprising as explained above: the neuronal cultures used for the co-IP experiments were derived from mouse hippocampus whereas the co-IP experiments for the mouse brain were done for the whole mouse brain, which contains neurons from other brain regions as well as non-neuronal cells. Moreover, the difference could also come from the difference between *in-vitro* and *in-vivo* conditions. Still, the enriched GO terms of the transmembrane proteins identified from cultured mouse hippocampal neurons and from adult mouse brain showed a substantial overlap (new Supplementary Fig. 11). We also acknowledge that pooling together all the transmembrane proteins from the three co-IP experiments using α II-spectrin, β II-spectrin, or α -adducin as the bait would increase the numbers of falsely identified MPS-interacting transmembrane proteins as well as the corresponding enriched GO terms. We added this cautionary note in the revised manuscript (See Page 12, 1st paragraph).

2. Since all the candidates tested in this study show a periodic distribution, it would be good to see a negative control for STORM-based imaging, such a protein which are not associated with the MPS, and thus lack a periodic pattern of localization. This is really crucial since it will illustrate the specificity of tested candidates to the MPS in neurons.

Response: As the reviewer suggested, we included four negative controls in the new Supplementary Fig.5 of the revised manuscript, including two cytosolic proteins (GFP molecules in GFP-expressing neurons and α -actinin-1) and two transmembrane or membrane-associated proteins (glutamate metabotropic receptor 2/3 and Cholera toxin B (CTB) which is known to

specifically bind to ganglioside GM1 enriched in plasma membrane). None of these molecules showed periodic distributions in the axons or dendrites (Supplementary Fig. 5).

3. The first paragraph states: “Magnetic beads were coated with the antibody that can specifically bind to a bait protein previously known to be an MPS structural component, such as β II-spectrin, α II-spectrin and α -adducin, ...”. Later, however, the α II-spectrin is chosen as one of the five candidates for validation. Please clarify the rationale behind this. If the authors question the role of α II-spectrin in the MPS, why it was used in co-IP experiments?

Response: α II-spectrin and β II-spectrin form spectrin tetramers that are expected to be a structure component of the MPS, we hence expect α II-spectrin to exhibit periodic distributions in the axons and used α II-spectrin as a bait. α II-spectrin has also been previously shown to be periodically distributed at the axon initial segment (AIS) [Huang et al, J. Neurosci. (2017)]. In this work, we verified that α II-spectrin is indeed also a MPS structural component and shows a periodic distribution in axonal regions outside the AIS. We clarified this now in our revised manuscript (Page 7, 2nd paragraph).

4. The last paragraph states: “ β II-spectrin or ankyrin B knockdown only led to a small reduction in the cell-surface expression levels of L1CAM, but did not decrease the cell-surface expression levels of NCAM1 or CHL1 (Supplementary Fig. 8).” But the data in Fig. S8 show a significant increase in surface levels of NCAM1 and CHL1, or am mistaken? In case the graph is correct, could the increase in the surface levels of NCAM1 and CHL1 be the cause of alterations in the neurite bundling and synapse density?

Response: We performed this control experiment to measure the cell-surface expression levels of the cell adhesion molecules because we wondered whether the observed reduction in axon-axon and axon-dendrite interactions could be simply explained by a reduction in the cell-surface expression levels of these cell adhesion molecules in the β II-spectrin knockdown or ankyrin B knockdown neurons. Supplementary Fig. 14 (old Supplementary Fig. 8) shows a moderated increase in the surface expression levels of NCAM1 and CHL1 in β II-spectrin or ankyrin B knockdown neurons. Although it is formally possible, we think it is unlikely that a moderate increase in the expression levels of NCAM1 and CHL1 would cause the substantial reduction in neurite bundling and synapse density observed in the β II-spectrin or ankyrin B knockdown neurons. We now clarify this point in our revised manuscript (Page 15, 2nd paragraph).

Minor:

1. Please describe the criteria, which were used to choose the validation targets (α II-spectrin, tropomodulin 1, tropomodulin 2, dematin and coronin 2B) from the list of 515 immunoprecipitated proteins?

Response: Among the immunoprecipitated proteins, we made the priority to validate the immunoprecipitated proteins that are previously known as actin or β II-spectrin binding proteins. We added this description to our revised manuscript (Page 7, 2nd paragraph).

2. I am missing the evaluation of β II-spectrin KD efficiency for analysis in Fig. 1d? KD efficiency is analyzed as a part of Fig. S5f, but this is not clear if these results also belong to Fig.1.

Response: We apologize for the confusion. The quantification of the β II-spectrin knockdown efficiency shown in the old Supplementary Fig. 5f was also for the multiplexed quantitative mass spec analysis based on the TMT isobaric labeling (old Fig. 1d; new Supplementary Fig. 3). To make this point clearer, we moved this knockdown efficiency quantification to Supplementary Fig. 3a and cited this quantification figure when we introduced the multiplexed quantitative mass spec analysis.

3. To verify the targets, the authors could also perform the IP and MS analysis under the β II-spectrin KD conditions.

Response: We thank the reviewer for this suggestion. We note that β II-spectrin knockdown condition may not be the best negative control because the knockdown efficiency of β II-spectrin using the β II-spectrin shRNA is ~70% and the co-IP using the β II-spectrin antibody may still pull down and enrich the remaining β II-spectrin in the neuronal compartments. Therefore, we performed two other co-IP experiments as the negative controls: First, we used a mouse IgG control antibody, which is not directed against any known antigen, to perform the co-IP and mass spec analysis (Supplementary Fig. 1). To remove the non-specifically bound proteins from the identified protein list, we used the distributed Normalized Spectral Abundance Factor (dNSAF) previously reported as a label free quantitative measure of protein abundance [Zhang et al, Anal Chem 82, 2272-2281 (2010)]. For each co-IP experiment (i.e, using α II-spectrin, β II-spectrin, or α -adducin as the bait), we calculated the fold-changes of dNSAF values in the co-IP experiment with α II-spectrin, β II-spectrin, or α -adducin as the bait vs the IgG negative control, and removed the proteins whose dNSAF fold-change value were not greater than 1. The mass spec analysis results for these control experiments are provided in the revised Supplementary Table 1. Second, we also performed co-IP under the condition of LatA and CytoD treatment, which is known to disrupt the MPS, which also showed drastic reduction in co-immunoprecipitated proteins (Supplementary Fig. 1).

4. The MPS structure is disrupted by the knockdown of several MPS candidates validated by the average autocorrelation amplitude of β II-spectrin (Fig. 2f) leading to increased axon diameter. Is the diameter of the dendrites also increased or is this phenotype specific to axons? Do the neurons show an altered morphology / complexity?

Response: As suggested by the reviewer, we compared the dendrite diameters for β II-spectrin-knockdown neurons and adducin-knockdown neurons to the results measured on neurons transfected with control (scrambled) shRNA. We indeed observed a moderate increase in the dendrite diameter as well (new Supplementary Fig. 8).

5. In Fig. S1a & 1b please indicate the input amount of proteins that were given to the antibody-coupled beads from brain lysates and lysed cultured neurons.

Response: We have added the input protein amounts for the co-IP experiments in the legend of Supplementary Fig. 1.

6. I am missing the number (n) of axons analyzed per experiment in Fig. 4.

Response: We have included the number of axon segments analyzed in the legend of Fig. 4 as well as in other related figures.

7. Please add an example illustrating KD experiments in Suppl. Figure 4a.

Response: We have added the example images to illustrate the KD efficiency in the new Supplementary Fig. 7a (old Supplementary Fig. 4a).

8. For Fig. S8 please indicate the p-values in b and c, as well as the experimental N.

Response: We have included the p-values and the number of experiment n to new Supplementary Fig. 14 (old supplementary Fig. 8).

9. Discussion part is too long and can be shortened by half.

Response: We thank the reviewer for the suggestion. There was indeed some redundancy between the Results and Discussion sections. We have revised and shortened the Discussion part.

Reviewer #2 (Remarks to the Author):

The manuscript shows evidence for proteome in neurons interacting directly or indirectly with proteins known to be main components of the MPS. To do so, mass spectrometry is conducted in protein pull-down obtained from 4 independent co-IP experiments: samples obtained from co-IP experiments against α 2-spectrin, β 2-spectrin or alpha adducin applied in mouse whole brain lysates and against β 2-spectrin applied in 20DIV hippocampal neuronal cultures. To narrow down MPS-interacting candidate proteins, authors consider those which are present in all 4 co-IP experiments, arriving to a list of 515 protein candidates.

The manuscript then evaluate some specific proteins by super-resolution microscopy to determine if these organize periodically in axons from hippocampal neurons in culture, in a way consistent with the MPS (190nm lags). These include known actin-interacting proteins and also signaling proteins.

The manuscript then evaluates some other candidates for their impact their knock-down has in the MPS structure, as seen by β 2-spectrin STORM imaging, providing evidence for proteins involved in MPS structure itself.

Finally, authors examine more specific protein families or groups for their biological function by loss-of-function approaches. Authors arrive to the conclusion that non-muscle myosin II (NMII) function within a same actin "ring" to maintain axon diameter, a series of MPS-interacting transmembrane adhesion proteins function to allow a normal interaction of the axon to other axons or dendrites.

General concerns:

- The manuscript is weak in building upon a vast existing literature that can be used to enrich the interpretation of their own results: These include works describing the axonal proteome (Zappulo et al 2017.) and previous direct protein-protein interaction biochemical evidence. Zappulo A, Bruck D Van Den, Ciolli Mattioli C, Franke V, Imami K, McShane E, Moreno-Estelles M, Calviello L, Filipchuk A, Peguero-Sanchez E, Müller T, Woehler A, et al. 2017. RNA localization is a key determinant of neurite-enriched proteome. Nat Commun 8:583.

Response: We thank the reviewer for this suggestion, but we feel that a comparison with the results in the Zappulo et al 2017 paper is unlikely to be very informative. In this work, we are interested in identifying proteins that interact with the MPS structure in neurons, whereas the Zappulo et al paper describes the proteome enriched in axons and dendrites, and the role of RNA localization in the neurite-enrichment of proteins. The neurite-enriched proteins described in Zappulo et al do not necessarily interact with the MPS.

- The manuscript is weak in organizing the vast evidence obtained into a proposal of a way in organizing a model of the MPS. For instance, others have tried to do so with the existing evidence and suggested organizing MPS-related proteins into "structural components" (those than when knocked down disrupt the MPS), proteins that directly interacts with structural components (but not substantially altering the MPS if absent), and proteins that will evidence a periodic distribution but not directly associated with the MPS. It is nothing else than trying to agree on what IS the MPS and what is ASSOCIATED with it. This will avoid confusions when using the term MPS. These reviews tried to organize this information: Leterrier et al 2017 and Unsain et al 2018

Leterrier C, Dubey P, Roy S. The nano-architecture of the axonal cytoskeleton. Nat Rev

Neurosci. 2017 Dec;18(12):713-726. doi: 10.1038/nrn.2017.129. Epub 2017 Nov 3. PMID: 29097785.

Unsain N, Stefani FD, Cáceres A. The Actin/Spectrin Membrane-Associated Periodic Skeleton in Neurons. Front Synaptic Neurosci. 2018 May 23;10:10. doi: 10.3389/fnsyn.2018.00010. PMID: 29875650; PMCID: PMC5974029.

Response: We thank the reviewer for pointing out two previous reviews on the neuronal MPS structures. We now cite these review papers in our Discussion part in the revised manuscript (Refs. 48 and 49). The reviewer suggested to organize MPS-related proteins into i) “structural components” (those that when knocked down disrupt the MPS), ii) proteins that directly interacts with structural components (but not substantially altering the MPS if absent), and iii) proteins that will evidence a periodic distribution but not directly associated with the MPS. We expanded the structural components to include some proteins that are part of the MPS structure but are not required for the formation of the MPS and hence knockdown of the protein does not disrupt MPS. For example, like tropomodulin1, tropomodulin 2 is an actin-binding protein that caps the actin filaments and is present in the MPS structure, even though knocking down tropomodulin 1 but not tropomodulin 2 led to disruption of the MPS. We thus included tropomodulin 2 as a structural component as well. For the two additional categories (ii and iii) suggested by the reviewer, proteins that directly interact with structural components, and proteins that will evidence a periodic distribution but not directly associated with the MPS, we did not differentiate these two categories in our paper, but rather considered proteins that are either directly or indirectly associated with the MPS both as proteins interacting with the MPS. We now clarified in our revised manuscript that by MPS-interacting proteins, we are referring to proteins either directly or indirectly interacting with the MPS (Page 5, 1st paragraph). We also note that most of the candidate MPS-interacting proteins that we identified by our co-immunoprecipitation and mass spec analysis are not previously known MPS-interacting proteins. Although many of these candidate proteins need to be validated with additional experiments, as is generally true for the interactomes identified by co-immunoprecipitation and mass spec, these candidate MPS-interacting proteins provide interesting hypotheses for future studies.

- STORM imaging as a validation for protein-protein interactions is overstated.

Throughout the text, authors state that STORM imaging was used to validate interactions first obtained by the co-IP experiments. I do not agree with that. An experiment to validate interactions typically would look for interactions or close proximity by other means, like FRET, Bimolecular Fluorescence Complementation Analysis, Surface plasmon resonance, etc. Using super-resolution microscopy would be useful if co-detections are made and it is clearly calculated for each experiment what is the resolution attained, and then one can say proteins A and B are present in the same volume of aprox. 30-60 nm xy. In summary, evidence presented by STORM imaging show that the candidate proteins show a periodic distribution consistent with the MPS, but is not a validation of interaction. Also, being a work mainly based on protein-protein interactions, it is striking that are virtually no co-immuno-fluorescence detection using STORM, and not at all imaging for filamentous actin, which is a central player in the MPS and the focus of figure 2. I strongly suggest these validation claims are changed and that evidence from co-immunolabeling STORM with the newly discovered MPS-related proteins is included, to determine their position with respect to structural components of the MPS (namely filamentous actin and bil-spectrin).

Response: We believe that this comment is due to a misunderstanding, and we apologize if the statements in our original manuscript led to this misunderstanding. In this manuscript, “interaction” is used to refer to both direct and indirect interactions. Likewise, we used the phrase “MPS-interacting proteins” to refer to the proteins that are either directly interacting with the MPS or indirectly associated with the MPS through other intermediate molecules. We now

clarify this point explicitly in the revised manuscript (Page 5, 1st paragraph). When a protein shows a periodic distribution with the same periodicity as observed for spectrin and actin, we believe that it is reasonable to suggest that this protein is either directly or indirectly associated with the MPS structure.

In addition, as suggested by the reviewer, we performed the two-color STORM (co-immunolabeling STORM) imaging experiments for the validated proteins in Fig. 2. α II-spectrin was co-imaged with adducin; tropomodulin 1, tropomodulin 2, dematin, and coronin-2B were co-imaged with β II-spectrin. Since adducin is an actin capping protein which has been shown previously to exhibit a periodic distribution and colocalize with actin (Xu et al, Science 2013), the locations of the periodic adducin stripes indicate where the actin rings are located. Since the immuno-labeling site of the β II-spectrin is near the center of the spectrin tetramer, the locations of the periodic β II-spectrin stripes should be mid-way between two adjacent actin rings. Our two-color STORM data are thus consistent with the notions that α II-spectrin forms spectrin tetramers with β II-spectrin to connect the adjacent actin rings, and that tropomodulin 1, tropomodulin 2, dematin, and coronin-2B are colocalized with the actin rings. These new two-color STORM imaging results are shown in the new Supplementary Fig. 6.

Particulars

-Page 3, Paragraph 1. Author's lists proteins as MPS "interacting proteins", but for most of them interaction evidence was lacking, albeit have been shown to exhibit a periodic organization... but do not necessarily interact with the MPS. Again to do such a claim it is necessary that authors define more clearly what the MPS is.

Same comment can be done in Page 11. "...previously shown to bind to the MPS and exhibit periodic distributions...".

Again, the interpretation of previous work is overstated. In the literature, evidence of binding of these proteins to the MPS is mostly lacking. A protein can show MPS-induced periodic distribution without directly binding to its structural components. I believe that given the main goal of the present report, authors should be more specific about the evidence from the literature and explain in detail each of the possibilities.

Response: As noted in our response to an earlier comment from the reviewer, in this manuscript, "MPS-interacting proteins" refer to the proteins that either directly associated with the MPS or indirectly associated with the MPS through other intermediate molecules. We now clarify this point explicitly in the revised manuscript (Page 5, 1st paragraph).

-Page 4, 2nd paragraph, "Furthermore, treatment with actin disrupting drugs (LatA and CytoD), which are known to disrupt the MPS"... What was treated, the neurons in culture or the lysates, or the pull-downs?, How? For how long? Dose? For the little information provided I interpret as being added to the protein lysates... By the way these drugs affect the dynamic turnover of actin filaments I don't see how these can even be effective in a cell-free system. Please clarify the details of the experimental approach and then its interpretation.

Also, for a completeness of the interpretation of the data, authors must mention that LatA and CytoD will disrupt the MPS BUT ALSO MANY other actin-related biological processes such as endo- and exocytosis, Golgi apparatus dynamics, vesicle transport to the vicinity of the membrane, among others.

Response: For the co-IP experiment using the mouse brain tissues, F-actin disrupting drugs (20 μ M LatA and 40 μ M CytoD) were added to the whole brain lysate prior to incubating the brain lysate with the beads coated with the β II-spectrin antibody, and the F-actin disrupting drugs remained in the lysate during the entire period of this incubation step. For the co-IP experiment

using the mouse cultured hippocampal neurons, cultured neurons were incubated with 20 μ M LatA and 40 μ M CytoD for 2 hours before the cultured neurons were lysed and 20 μ M LatA and 40 μ M CytoD were also added to the culture neuron lysate as described for the brain tissue lysate. We have now included this information in our revised manuscript (Page 24, 5th paragraph).

LatA and CytoD inhibit actin polymerization either by sequestering actin monomers or by binding to the barbed end of actin filaments. Both sequestering actin monomers and binding to the barbed end of F-actin can happen in live cells or cell lysates. In addition, although LatA and CytoD may also disrupt other F-actin-related biological processes, disrupting other F-actin-related biological processes should not affect the interpretation of our data significantly. We used these co-IP experiments under the F-actin disrupting drug treatment as one of the negative controls to support that the proteins co-immunoprecipitated with β II-spectrin were pulled down by the β II-spectrin antibody through F-actin-dependent interactions. We have also been very careful in the interpretation of our co-IP experiments and stated that the proteins pulled down by the bait (α II-spectrin, β II-spectrin, or α -adducin) were potential candidate MPS-interacting proteins that need further validation and that by MPS-interacting proteins, we referred to proteins that interact either directly or indirectly with the MPS (see such cautionary statements in Page 4, last paragraph, Page 5, 1st paragraph, and Page 6, 2nd paragraph).

-On the right side of supp fig 1 a and b, the names of certain proteins appear. The IgG can be seen by one of the lanes, but all the others are only estimates, since the stain cannot pick up individual proteins. Please clarify that or I would suggest to change labels or identify individual proteins by immunolabeling. There are good arguments to pick up some visible bands as being actin and spectrin or IgG chains... but I don't agree that is true for alpha-adducin.

Response: We thank the reviewer for the suggestion. In the revised manuscript, we have clarified this by stating that these labels next to the gel images indicate the positions for the expected weights of spectrin (~280 kDa), α -adducin (~103 kDa), actin (~42 kDa), IgG heavy chain (~50 kDa), and IgG light chain (~ 25 kDa), rather than indicating the proteins themselves. We also note that there is a visible gel band at each of the above five positions in the SDS-PAGE images of co-immunoprecipitated proteins, and the gel bands at 280 kDa, 103 kDa and 42 kDa all showed an enrichment compared to the SDS-PAGE images of the co-immunoprecipitated proteins in control pulldown experiments using irrelevant IgG or under LatA and CytoD condition that disrupt the MPS, indicating that these three gel bands are likely spectrin, α -adducin, and actin. But as suggested by the reviewer, to be more cautious, we stated in the caption of Supplementary Fig. 1 of the revised manuscript that the labels next to the gel image indicate the positions of the expected weights of the proteins without specifically claiming certain gel bands must represent spectrin, α -adducin, or actin.

-Page 4, 3rd paragraph, spelling: "...tissues, i) using α -adducin as..." should say iii).

Response: This sentence has been revised.

-"...of the MPS structures, for both 1D and 2D forms, in axons, dendrites and soma..." In the literature, no one uses the term MPS to refer to the topology of these components in the soma, referred by the authors as 2D. If that was the case, then erythrocytes would have an MPS. Also, the "abundance" of these 2D topologies in somas is very low. I suggest authors just acknowledge that they might be getting interactors that are not present in the MPS of axons and dendrites, since they have lysed the whole cell.

Response: When we reported the 2D polygonal structure in the neuronal cell body in our 2017 PNAS paper, we referred to this structure as the 2D MPS to acknowledge the fact that the

structure has similar molecular components as the 1D MPS structure in axons and dendrites. While we consider the terminology 2D MPS reasonable, we revised the text in the manuscript according to the reviewer's suggestion and avoided using the term MPS to describe the membrane skeleton structure in the somas of neurons.

-In general, the results description should include some key experimental details for an easier appreciation of the results. For example, to appreciate the quantitative mass-spec upon beta spec knockdown, it is key to know the timing of the knockdown and subsequent analyses. Most interventions lack key information more at hand in the Results section. The Mat and Met section also lacks that information. I strongly suggest that for every shRNA experiment or OE or drugs treatment, authors identify WHEN it is treated (be it shRNA or drugs) and when samples are collected or fixed. This is very important for a correct interpretation of the data, especially to acknowledge that if interventions of cells in culture are performed early, they might be affecting formation of the MPS, while when treatments are much later, they might be affecting maintenance of it.

Response: As suggested by the reviewer, we have added this information in the Methods section (Page 22, 1st and 2nd paragraphs; Page 23, 4th paragraph; Page 24, 4th – 7th paragraphs; Page 25, 2nd – 4th paragraphs).

-When comparing antibodies, it should be specified whether they target the same or different epitopes in the target protein, since that can be a source of disruption of the periodic visualization. This will be fair to acknowledge. Some proteins have long moving tails that even if well detected will not show a periodic structure, as is the case for the N- and C- terminus of Ankyrins.

Response: As suggested by the reviewer, we added the epitope information in the figure captions of Fig. 2, Fig. 4, new Supplementary Fig. 4c, and new Supplementary Fig. 6.

-Figure 2. If actin binding partners are being tested by STORM, why aren't they compared directly to actin distribution? Or at least shown in the same set of experiments? Or compared that they are off-phase compared to beta spec? This is important since authors then build up a model (Fig 6) based mostly on previous knowledge, but not from a direct observation within the MPS. i.e. What is their evidence for tropomodulin or dematin to interact with the actin rings?

Response: As suggested by the reviewer, we performed two-color STORM imaging for the five structural components (α II-spectrin, tropomodulin-1, tropomodulin-2, dematin, coronin-2B) shown in Fig. 2. α II-spectrin was co-imaged with adducin; tropomodulin 1, tropomodulin 2, dematin, and coronin-2B were co-imaged with β II-spectrin. These new two-color STORM experiments are shown in the new Supplementary Fig. 6. Since adducin is an actin-capping protein which has been shown previously to exhibit a periodic distribution and colocalize with actin (Xu et al, Science 2013), the locations of the periodic adducin stripes indicate where actin rings are located. Since the immuno-labeling site of the β II-spectrin is near the center of the spectrin tetramer, the locations of the periodic β II-spectrin stripes should be mid-way between two adjacent actin rings. Our new two-color STORM data are thus consistent with the notions that α II-spectrin forms spectrin tetramers with β II-spectrin to connect the adjacent actin rings, and that tropomodulin 1, tropomodulin 2, dematin, and coronin-2B are colocalized with the actin rings.

-Figure 2, Experiment with PH and CH deletions. To assess if b2-spec deletions really disturb the MPS, STORM imaging should be performed with at least one other structural component of the MPS, apart from b2-spec (actin, α 2-spec, tropomodulin 1, dematin, etc). Is it the MPS itself disrupted or is it the way (or capability) of truncated b2-spec to be incorporated in the MPS that

is disrupted?

Response: This comment seems to be caused by a misunderstanding that we expressed truncated β II-spectrin mutants on the background of wildtype neurons where endogenous β II-spectrin is present. In fact, we expressed truncated β II-spectrin mutants on the background of β II-spectrin knockout neurons. Because these truncated β II-spectrin mutants did not adopt a periodic structure, we concluded that these truncated β II-spectrin mutants cannot form the MPS.

In light of the comment from the reviewer, we performed new imaging experiments for these truncated β II-spectrin mutants on the background of wildtype neurons expressing β II-spectrin (see new Supplementary Fig. 9). We found that these truncated β II-spectrin mutants (both PH domain and CH domain deletions) overexpressed on the background of wildtype neurons did not show periodic distribution. On the other hand, when we introduced the overexpression of full-length β II-spectrin in wildtype neurons in the same way as we overexpressed the β II-spectrin mutants, the overexpressed full-length β II-spectrin did exhibit a periodic distribution. We also imaged endogenous adducin in these neurons, which still showed a periodic distribution when either full length β II-spectrin or truncated β II-spectrin mutants were overexpressed, suggesting that the overexpression of truncated β II-spectrin mutants did not significantly affect the endogenous MPS structures.

Overall, all our experiments support the notion that these truncated β II-spectrin mutants (both PH domain and CH domain deletions) cannot be incorporated into the MPS structure.

Reviewer #3 (Remarks to the Author):

The manuscript by Zhou and co-authors reports the membrane-associated periodic skeleton (MPS) proteome in neurons, and performs high-resolution microscopy and functional analysis to validate some of the new MPS components in neurons.

I have a major methodological concern on the proteomics data and how it is reported. Particularly with the experiments directed to identify MPS neuronal components (Fig. 1), which represent the foundations of the rest of the manuscript. Overall the detail on the MS methods used is insufficient (i.e. I could not find information on the number of replicates performed, are there biological or technical replicates ?) . Particularly concerning is how the authors dealt with proteins identified in the negative control IPs. Actually, it is unclear to this reviewer if they performed MS analysis on the negative control samples. I could not find in the manuscript or the supplementary tables the proteins identified in the negative control.

Response: We thank the reviewer for these suggestions to improve the description of our mass spec results and to improve our mass spec analysis. We addressed the reviewer's concerns by providing the requested information: For Fig. 1b (including the co-IP experiments from the adult mouse brain or from cultured hippocampal neurons using a known MPS component as the bait), each co-IP condition contains two biological replicates. For old Fig.1d (new Supplementary Fig. 3; comparing the proteomes in the wild-type neurons and in the β II-spectrin knockdown neurons), each condition contains three biological replicates. We now add this information to the captions of Fig. 1 and Supplementary Fig. 3.

In addition, we performed new mass spec analysis on the negative control samples as suggested by the reviewer (see below for details).

Analysing the negative control is key in order to define the true partners of β II-spectrin, α II-spectrin and α -adducin. The authors must analyse the negative control (if they

haven't) and report the proteins therein. Finally, they must explain how they subtract the data from the negative controls from the data in the problem IPs. Spectrins are very abundant proteins and I wouldn't be surprised if there are traces of these proteins found in the negative control IP. In this case the authors should use a quantitative criterion to remove proteins identified in the negative control from those present in the problem IPs. If this has been done it should be properly reported. If it hasn't been done it should be done as this is key to remove false positives from problem IPs. The set of 515 MPS components would not be trust-worthy if the proteins in the negative control have not been removed from those in the problem IPs.

The authors present SDS-PAGE images of IP'ed samples, including the negative control. There it is clear that the negative control pulls-down far less proteins than problem IPs (Suppl Fig 1), indicating that the IPs are selective. Yet, a clear protein staining is visible in the negative control lanes, indicating that there are proteins spuriously IPed. This would be totally expected. It is a limitation of IPs and, in general, of affinity purification procedures. As mass spectrometry is very sensitive, many proteins will be identified in the negative control. Again, the authors should find a way to control for this.

Response: We thank the reviewer for the suggestion to more quantitatively analyze the negative control samples where an irrelevant IgG antibody not directed against any known antigen was used to pull down proteins nonspecifically. In our original manuscript, we only presented the SDS-PAGE images of these negative control samples and did not present the mass spec analysis for them. In the revised manuscript, we presented the mass spec analysis results for the negative controls as suggested by the reviewer. We then used the distributed Normalized Spectral Abundance Factor (dNSAF) previously reported as a label free quantitative measure of protein abundance [Zhang et al, *Anal Chem* **82**, 2272-2281 (2010)]. As suggested by the reviewer, for each co-IP experiment (using α -spectrin, β II-spectrin, or α -adducin as the bait), we calculated the fold-changes of dNSAF values in the co-IP experiment using the proper baits versus the negative control using the irrelevant IgG, and defined the non-specifically bound proteins as the proteins with the dNSAF fold-changes that were not greater than 1. We removed these non-specifically bound proteins from our reported identified protein list. The new mass spec analysis results for these control experiments are provided in the revised Supplementary Table 1.

From a formal stand point the results sections results very long (12 pages), including many elements of discussion. As a result the discussion is a bit redundant. Results could be written in a more succinct manner, leaving the discussion aspects for the final section of the manuscript.

Response: We thank the reviewer for the suggestion and have revised the manuscript to reduce the redundancy.

Minor comment:

In the primary cultures, did the authors use drugs to prevent glial (or other mitotic cells growth?). This is quite standard in the field and it would be particularly important in this manuscript, as they aim at identifying the neuronal MPS proteome (not the glia MPS proteome).

Response: In our primary neuronal cultures, AraC (2 μ M) was added to the culture medium at DIV 2 to prevent glial proliferation for all the neuronal cultures used for mass spec analyses. We have now clarified this in the revised manuscript (Page 20, 2nd paragraph).

Proteomics data should be made available through public repositories (i.e. PRIDE) and the corresponding link provided in the manuscript.

Response: We have submitted the proteomics data to the ProteomeXchange Consortium via the PRIDE partner repository with the data set identifier PXD030886 (<https://www.ebi.ac.uk/pride/archive/projects/PXD030886/private>). The data will be made publicly available upon the publication of the paper, and currently the submitted data can be accessed using the reviewer account below:
Username: reviewer_pxd030886@ebi.ac.uk
Password: uA8srs8N

REVIEWERS' COMMENTS

Reviewer #1 (Remarks to the Author):

The authors addressed most of my major concerns. Updated version became more transparent with respect of the data analysis (including the uploading the MS data to the PRIDE database). I appreciate the addition of negative control samples to strengthen the main message of the manuscript. The authors also included the number of performed experiments in corresponding figure legends, even though I am still concerned about the low number of biological replicates for the immunoprecipitation experiments in Fig. 1b (two biological replicates).

Including the data for negative control samples gives the study more credibility. The manuscript also contains now the comprehensive description of the MS data processing. I also appreciate the application of the chosen threshold (fold-change 1) on the dNSAF values, which makes the identification of binding partners more reliable.

Reviewer #2 (Remarks to the Author):

After carefully reading the rebuttal letter (from all 3 reviewers), the revised manuscript, their new experiments and conclusions, all my previous comments and concerns were addressed and I have no more comments or concerns to add.

I think the revisions have considerably strengthened their conclusions and quality of the data. The work will be very valuable for the research community working in this area.

Sincerely, Nicolas Unsain

Reviewer #3 (Remarks to the Author):

The authors have thoroughly addressed my technical comments on the MS data, especially regarding the use of a negative control. Although it still results shocking that they did not include a negative control in their initial version of the manuscript, the data would now be suitable for publication.

Furthermore, the manuscript is now more clearly written.

In my opinion this manuscript presents relevant new insights on the biology of MPS in neurons and will likely be a relevant resource for the community.

Response to reviewer comments

Reviewer #1 (Remarks to the Author):

The authors addressed most of my major concerns. Updated version became more transparent with respect of the data analysis (including the uploading the MS data to the PRIDE database). I appreciate the addition of negative control samples to strengthen the main message of the manuscript. The authors also included the number of performed experiments in corresponding figure legends, even though I am still concerned about the low number of biological replicates for the immunoprecipitation experiments in Fig. 1b (two biological replicates). Including the data for negative control samples gives the study more credibility. The manuscript also contains now the comprehensive description of the MS data processing. I also appreciate the application of the chosen threshold (fold-change 1) on the dNSAF values, which makes the identification of binding partners more reliable.

Response: We thank the reviewer for appreciating that although the biological replicate number was two per condition for the co-IP experiments in Fig. 1b, our inclusion of data for negative control samples gives the study more credibility and that our application of a threshold on the dNSAF fold change between capture co-IP and negative control co-IP experiments makes the identification of binding partners more reliable.

We additionally note that although only two biological replicates were performed for each co-IP bait protein (α -spectrin, β II-spectrin, and α -adducin), the proteins identified as candidate MPS-interacting proteins in cultured hippocampal neurons and adult mouse brains were the proteins commonly identified using all three different baits (α -spectrin, β II-spectrin, and α -adducin). This means the identified proteins in Fig. 1b was based on six co-IP experiments (three baits and two replicate experiments for each bait) each for cultured neurons and for adult mouse brains, which should provide reasonable accuracy in determining the candidate MPS-interacting proteins. In addition, we have made it clear in our manuscript that these proteins are potential candidate MPS-interacting proteins and that further validation is required to confirm the interactions between the MPS and these mass spec-identified candidate proteins. Indeed, we have performed independent imaging experiments to validate a subset of these candidate MPS-interacting proteins.

Reviewer #2 (Remarks to the Author):

After carefully reading the rebuttal letter (from all 3 reviewers), the revised manuscript, their new experiments and conclusions, all my previous comments and concerns were addressed and I have no more comments or concerns to add. I think the revisions have considerably strengthened their conclusions and quality of the data. The work will be very valuable for the research community working in this area. Sincerely, Nicolas Unsain

Response: We thank the reviewer for his positive comments about our revised manuscript.

Reviewer #3 (Remarks to the Author):

The authors have thoroughly addressed my technical comments on the MS data, especially regarding the use of a negative control. Although it still results shocking that they did not include a negative control in their initial version of the manuscript, the data would now be suitable for publication. Furthermore, the manuscript is now more clearly written.

Response: We thank the reviewer for his/her positive comments about our revised manuscript.